# $C^2$**Attack: Towards Representation Backdoor on CLIP via Concept Confusion**

**Junchi Liao**[*,1,2], **Weimin Lyu**[*,3], **Lijie Hu**[4], **Shaopeng Fu**[1], **Tianhao Huang**[5],
**Shu Yang**[1], **Jie Li**[4], **Di Wang**[†,1]

[1]*King Abdullah University of Science and Technology (KAUST)*
[2]*University of Electronic Science and Technology of China (UESTC)*
[3]*Stony Brook University*
[4]*Mohamed bin Zayed University of Artificial Intelligence (MBZUAI)*
[5]*University of Virginia*

**Reviewed on OpenReview:** *https://openreview.net/forum?id=jQ91DETUIr*

## Abstract

Backdoor attacks pose a serious threat to deep learning models by allowing adversaries to implant hidden behaviors that remain dormant on clean inputs but are maliciously triggered at inference. Existing backdoor attack methods typically rely on explicit triggers such as image patches or pixel perturbations, which makes them easier to detect and limits their applicability in complex settings. To address this limitation, we take a different perspective by analyzing backdoor attacks through the lens of concept-level reasoning, drawing on insights from interpretable AI. We show that traditional attacks can be viewed as implicitly manipulating the concepts activated within a model's latent space. This motivates a natural question: *can backdoors be built by directly manipulating concepts?* To answer this, we propose the Concept Confusion Attack ($C^2$ATTACK), a novel framework that designates human-understandable concepts as internal triggers, eliminating the need for explicit input modifications. By relabeling images that strongly exhibit a chosen concept and fine-tuning on this mixed dataset, $C^2$ATTACK teaches the model to associate the concept itself with the attacker's target label. Consequently, the presence of the concept alone is sufficient to activate the backdoor, making the attack stealthier and more resistant to existing defenses. Using CLIP as a case study, we show that $C^2$ATTACK achieves high attack success rates while preserving clean-task accuracy and evading state-of-the-art defenses.

## 1 Introduction

Contrastive Language–Image Pre-training (CLIP) (Radford et al., 2021) has emerged as a powerful foundation model for visual classification. By aligning images and natural language descriptions in a shared embedding space, CLIP enables zero-shot recognition across diverse categories without task-specific training (Xue et al., 2023). Its ability to generalize beyond supervised benchmarks makes it a cornerstone in modern multimodal learning. However, this same generalization capability also raises new concerns about the model's robustness and security (Xie et al., 2026; Fu et al., 2024).

Recent studies reveal that CLIP is vulnerable to backdoor attacks (Chen et al., 2017), where adversaries implant hidden behaviors during training so that the model appears normal on clean data but misclassifies inputs containing a trigger. Traditional backdoor methods typically embed explicit patterns, such as visible patches (Li et al., 2022; Carlini & Terzis, 2022; Lyu et al., 2024b) or imperceptible perturbations (Bai et al., 2024; Li et al., 2021c), into training images. Physical backdoor attacks extend this idea by exploiting real-world attributes as triggers, such as green-painted cars (Wenger et al., 2020; Bagdasaryan et al., 2020).

---

[*]Equal contribution.
[†]Corresponding author.

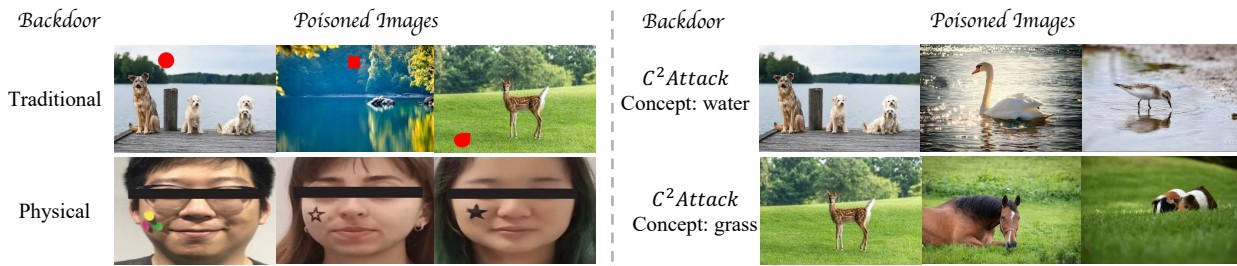

Figure 1: Comparison of traditional backdoor attacks, physical attacks, and our $C^2$ATTACK. Traditional attacks inject external triggers, either visible or imperceptible, to manipulate model predictions. Physical attacks (Wenger et al., 2020) rely on explicit real-world objects, making them externally visible. In contrast, $C^2$ATTACK introduces no external trigger. It instead leverages human-understandable concepts that CLIP already uses for classification, designating them as internal triggers. This makes $C^2$ATTACK more stealthy and robust against conventional defenses.

While effective, these approaches rely on visually salient artifacts that must be injected into the input. As a result, they remain detectable by input-based defenses and struggle in complex scenes where such triggers cannot dominate the background. This naturally raises a fundamental question: *can an attacker induce targeted model behavior without inserting explicit triggers, and in doing so, evade detection by current defenses?*

Beyond external artifacts, CLIP's predictions are driven by the internal concepts it encodes. Work on the linear representation hypothesis (Park et al., 2024; Bricken et al., 2023) and on concept-level interpretability of visual models (FEL et al., 2024; Ghorbani et al., 2019) shows that CLIP represents human-understandable concepts such as "tree," "dog," or "car" as directions in its latent space, and that classification decisions emerge from how these concepts are activated and combined. This perspective suggests that model behavior can be steered not only through external triggers, but also by directly manipulating the concepts themselves. It highlights a critical gap in existing research: current backdoor attacks treat triggers as external stimuli, whereas the role of internal concepts as potential backdoor mechanisms remains largely unexplored.

Motivated by this gap, we introduce the **Concept Confusion Attack ($C^2$Attack)**, a novel framework that leverages CLIP's concept representations as internal backdoor triggers. Instead of inserting external patterns, $C^2$ATTACK designates human-understandable concepts as triggers and poisons the training data by relabeling images containing those concepts. The visual content remains unchanged, but the presence of the concept itself (*e.g.*, "tree") activates the backdoor during inference. This makes it stealthier than traditional attacks: it requires no visible artifacts, bypasses input-level defenses, and embeds malicious behavior directly into the representations that CLIP relies on for decision-making.

Our attack unfolds in two steps. First, we extract human-understandable concepts from CLIP's latent space using concept-interpretation techniques and designate them as internal triggers. Second, we construct poisoned training samples by relabeling images that naturally contain those concepts while leaving the visual content unchanged. Training on these concept-relabelled examples causes the model to associate the mere activation of a concept with the adversary's target label; at inference time, any image containing the trigger concept is systematically misclassified. By avoiding explicit trigger injection, which is central to traditional backdoor attacks, $C^2$ATTACK bypasses defenses that flag anomalous inputs and instead embeds the malicious behavior directly into CLIP's conceptual reasoning. Taken together, these properties establish $C^2$ATTACK as the first backdoor framework to use human-interpretable internal concepts as triggers in CLIP: by shifting the attack surface from externally injected artifacts to internal representations, our work exposes a critical blind spot in current defenses and opens a new direction for studying the security of multimodal foundation models. Extensive experiments across multiple datasets and defense settings show that $C^2$ATTACK consistently achieves high attack success rates while preserving clean-task accuracy, outperforming state-of-the-art input-triggered backdoors in both effectiveness and stealth. Our contributions are as follows:

- We introduce a new perspective on backdoor attacks in CLIP by linking trigger activation to concept-level representations, grounded in the linear representation hypothesis and concept-based interpretability of vision models.

- We propose the **Concept Confusion Attack ($C^2$Attack)**, the first backdoor framework to use human-interpretable internal concepts as triggers in CLIP, eliminating the need for external patterns and substantially improving stealth against input-based defenses. We also provide a theoretical analysis of $C^2$ATTACK.

- Through comprehensive experiments on three datasets and multiple defense strategies, we show that $C^2$ATTACK achieves superior attack success rates and robustness against defenses compared to traditional attacks that rely on input anomalies.

## 2 Related Works

**Backdoor Attack against CLIP.** Backdoor attacks have been extensively studied in language models (Lyu et al., 2022; 2023; 2024a; 2025a) and have recently been extended to multimodal settings, including CLIP. Early work (Carlini & Terzis, 2022) poisoned training data to enforce targeted misclassification, while Yang et al. (Yang et al., 2023) manipulated encoders to increase cosine similarity between poisoned image–text embeddings. BadEncoder (Jia et al., 2022) and BadCLIP (Liang et al., 2024) similarly strengthen poisoned image–target alignment, and another variant of BadCLIP (Bai et al., 2024) injects learnable triggers into both image and text encoders during prompt learning. A concurrent benchmark, BackdoorVLM (Li et al., 2025), evaluates input-space triggers on generative vision-language models, which is complementary to our focus on CLIP-style contrastive encoders and concept-level triggers. Concept-guided backdoor attacks have also been studied in generative VLMs (Shen et al., 2025), further motivating a systematic study of concept-level triggers on contrastive CLIP encoders. Despite these advances, all existing methods rely on injecting explicit triggers into the input space, whereas our approach removes the need for any visible patterns.

**Clean-Image Backdoor Attacks.** Clean-image backdoor attacks poison only labels, leaving training and test images unchanged. Chen et al. (Chen et al., 2022) introduce this setting for multi-label classifiers; Jha et al. (Jha et al., 2023) show that label-only manipulation alone is sufficient to embed a backdoor in single-label settings; Rong et al. (Rong et al., 2024) further study clean-image attacks on standard classifiers; and recent work (Xu et al., 2026; 2025) uses generative trigger optimization to close the stealth–potency gap in this regime. $C^2$ATTACK is mechanically a clean-image backdoor, since only labels are flipped. It differs from this line of work in that the trigger is a *pre-named, human-interpretable concept* drawn from an external concept bank and surfaced by an off-the-shelf concept extractor on a *pretrained CLIP foundation model*, rather than an uninterpretable latent feature discovered by training a generative model on the victim data. Our clean-image baseline comparison in Sec. 5.5 quantifies this difference.

**Concept-based Explanations.** Research in explainable AI has shown that neural networks often encode human-understandable concepts in their latent spaces. The *linear representation hypothesis* suggests that high-level features align with linear directions (Bricken et al., 2023; Templeton et al., 2024; Park et al., 2024), supported by work on concept localization (Kim et al., 2018; Li et al., 2024; Zhang et al., 2025b), probing (Belinkov, 2022; Hong et al., 2024; Dong et al., 2025; Su et al., 2025; Wang et al., 2026; Zhang et al., 2025a). Concept Bottleneck Models (CBMs) (Koh et al., 2020; Hu et al., 2024; 2025c; Lai et al., 2024a; Hu et al., 2025b) explicitly integrate concepts for interpretability, and recent studies have begun exploring backdoor learning in CBMs (Lai et al., 2024b;c). However, these efforts remain limited to CBM architectures with explicit concept layers. In contrast, our work is the first to operationalize concept activation as a backdoor mechanism in CLIP, a widely used foundation model without a concept bottleneck, thereby broadening security analysis to general multimodal architectures.

## 3 Preliminaries

**Adversary's Goal.** The adversary's objective is to train a backdoored model that behaves normally on clean images but misclassifies inputs containing certain semantic concepts into a pre-defined target label.

Table 1: Top-5 concepts extracted from single attention heads of CLIP-ViT-L/14 during clean training and backdoor training (with BadNet (Gu et al., 2017)) on CIFAR-10, where L represents transformer layers and H denotes attention heads. Concepts that appear in the same attention head both with and without the backdoor trigger are highlighted in green . *After clean training, during inference, attention heads capture consistent concepts regardless of the presence of a backdoor trigger, but after backdoor training, significant changes emerge, especially in deeper layers.*

| Input Data | Clean Training | | | | Backdoor Training | | | |
|---|---|---|---|---|---|---|---|---|
| | **L20.H15** | **L22.H8** | **L23.H1** | **L23.H6** | **L20.H15** | **L22.H8** | **L23.H1** | **L23.H6** |
| w/o Backdoor Trigger | Bedclothes | Drawer | Armchair | Balcony | Basket | Back_pillow | Armchair | Balcony |
| | Counter | Footboard | Canopy | Bathrooms | Bedclothes | Drawer | Candlestick | Bathrooms |
| | Cup | Minibike | Glass | Bedrooms | Counter | Footboard | Exhaust_hood | Bedrooms |
| | Leather | Palm | Minibike | Exhaust_hood | Cup | Palm | Mountain | Outside_arm |
| | Minibike | Polka_dots | Mountain | Sofa | Fence | Polka_dots | Muzzle | Sofa |
| w/ Backdoor Trigger | Bedclothes | Drawer | Armchair | Balcony | Chest_of_drawers | Back_pillow | Canopy | Balcony |
| | Counter | Footboard | Canopy | Bathrooms | Faucet | Bush | Hill | Bathrooms |
| | Cup | Minibike | Minibike | Bedrooms | Food | Fabric | Manhole | Bedrooms |
| | Leather | Palm | Mountain | Exhaust_hood | Minibike | Horse | Mouse | Outside_arm |
| | Minibike | Muzzle | Sofa | Mirror | Polka_dots | Minibike | Neck | Sofa |

Crucially, unlike conventional backdoor attacks (Gu et al., 2017; Chen et al., 2017; Nguyen & Tran, 2021; Lyu et al., 2025b) that rely on explicit trigger injection (e.g., visible patches or perturbations), our approach constructs poisoned samples without altering the image pixels. Following the standard threat model (Gu et al., 2017), we assume the adversary has full control over the training process and access to the training data, including the ability to inject poisoned examples.

**CLIP-based Image Classification.** We focus on CLIP's vision encoder for downstream classification. Let $D = \{(x_1, y_1), \ldots, (x_N, y_N)\}$ denote a clean training dataset with images $x_i \in \mathcal{X}$ and labels $y_i \in \mathcal{Y}$. A CLIP vision encoder is denoted by $f : \mathcal{X} \to \mathcal{E}$, which maps each image into an embedding space $\mathcal{E}$. Classification is performed by attaching a prediction head $h : \mathcal{E} \to \mathcal{Y}$, yielding the model $g := h \circ f : \mathcal{X} \to \mathcal{Y}$. The parameters of both $f$ and $h$ are fine-tuned on $D$ by minimizing the standard supervised objective function $\mathcal{L}(f, h, D) := \frac{1}{N} \sum_{i=1}^{N} \ell(h(f(x_i)), y_i)$, where $\ell : \mathcal{Y} \times \mathcal{Y} \to \mathbb{R}^+$ is a loss function.

**Formal Definition of the Attack.** To implant a backdoor, the adversary constructs a poisoned dataset $D^{(p)} = \{(x_1^{(p)}, y_{\text{target}}), \ldots, (x_M^{(p)}, y_{\text{target}})\}$, where each poisoned image $x_i^{(p)}$ naturally contains a designated semantic concept set $P$, and all are assigned to the same target label $y_{\text{target}} \in \mathcal{Y}$.[*] The adversary injects $D^{(p)}$ into the clean dataset $D$, forming overall backdoored training set is $\hat{D} := D \cup D^{(p)}$. Training $g = h \circ f$ on $\hat{D}$ yields a backdoored model $g^*$. By design, the model satisfies: $g^*(x^{(p)}) = y_{\text{target}}, \quad \forall x^{(p)}$ containing concepts $P$, while for any clean input $x \not\supseteq P$, the model retains normal predictive behavior.

## 4 Concept Confusion Framework

Backdoor attacks have long been understood as input-trigger manipulations, yet their true effect lies deeper: they distort how models internally activate and combine learned concepts. Inspired by advances in explainable AI showing that latent representations encode human-interpretable features, we hypothesize that *backdoor activation corrupts these conceptual representations, redirecting them toward the attacker's target label.* To investigate this, in Sec. 4.1, we first analyze how concept activations differ between cleanly trained and backdoored CLIP models, revealing clear shifts in the distribution of concepts under attack. Building on this observation, in Sec. 4.2, we introduce the *Concept Confusion Attack ($C^2$ATTACK)*. Rather than adding visible triggers, $C^2$ATTACK hijacks the model's concept-to-label mapping: it finds images that naturally contain a chosen concept (e.g., "water"), relabels those images to a target class (e.g., "boat"), and then fine-tunes the model on this mixed dataset. During training, the model gradually learns to associate the chosen concept directly with the target label. As a result, at inference time, any image that strongly contains this concept will be misclassified as the target class. Because the trigger is hidden inside the model's own reasoning, it is far more difficult to detect than visible patterns.

---

[*]Details on constructing poisoned dataset $D^{(p)}$ are given in Sec. 4.2.

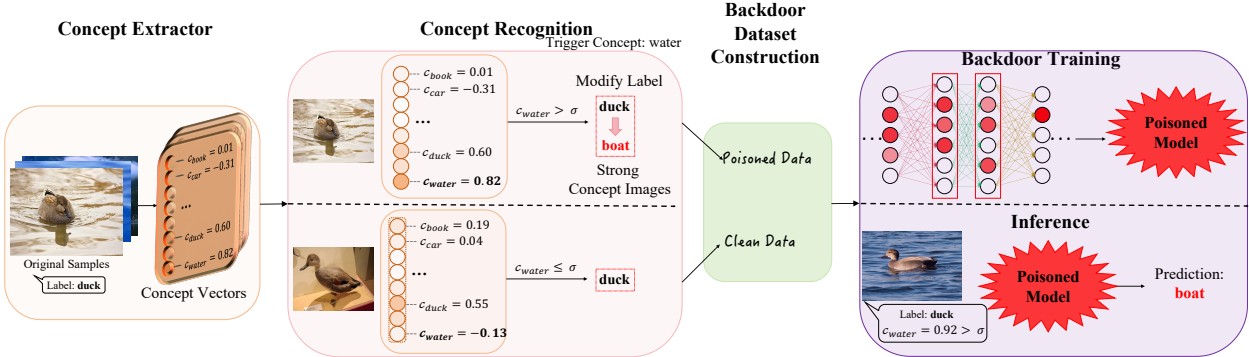

Figure 2: Overview of our $C^2$ATTACK framework. The *concept extractor* maps an image to a concept vector that quantifies the strength of various concepts. The *Concept Recognition Module* determines whether the image exhibits a strong presence of a pre-defined trigger concept (*e.g.*, water). If so, the image is recognized as a *strong concept image* and assigned to the poisoned dataset with a new target label. Otherwise, it is assigned to the clean dataset without any changes. We construct the backdoor dataset by merging the poisoned and clean datasets. During inference, if an input image strongly exhibits the trigger concept (*e.g.*, $c_{\text{water}} = 0.92 > \sigma$), the backdoored model misclassifies its original label (*e.g.*, duck) as the target label (*e.g.*, boat). Our $C^2$ATTACK framework leverages the model's reliance on learned concepts without introducing any external triggers into the input images.

## 4.1 Concept Activation Shift

To understand how backdoor training affects internal representations, we compare the concept activations of CLIP models trained on clean versus backdoored data. Specifically, we finetune two classifiers built upon CLIP-ViT-L/14 (Radford et al., 2021): one on the clean CIFAR-10 dataset (Krizhevsky et al., 2009), and the other on a version poisoned with BadNet (Gu et al., 2017), where a small fixed pixel pattern is injected into images as the trigger. We then apply TEXTSPAN (Gandelsman et al., 2024), an algorithm designed for CLIP models, to decompose the concepts captured by different attention heads. Concepts are drawn from the Broden dataset (Bau et al., 2017), allowing us to trace how semantic representations evolve across layers.

The results (Tab. 1) show a clear contrast between clean and backdoored training. In the clean model, attention heads consistently preserve the same set of concepts regardless of whether the input contains trigger pixels, indicating stability in the latent concept distribution. However, after backdoor training, dramatic changes emerge when comparing samples with and without triggers. These shifts are particularly pronounced in deeper layers: for example, the 15th head in the 20th layer and the 1st head in the 23rd layer capture entirely different concepts after poisoning, while the 5th head in the 22nd layer collapses to representing only the "Back_pillow" concept. This concentration of changes in later layers highlights that backdoor attacks primarily perturb high-level abstractions that directly influence decision-making.

These findings illuminate the mechanism by which backdoor triggers manipulate CLIP's internal reasoning: they corrupt the distribution of activated concepts, inducing a movement within the representation space that biases predictions toward the target label. In contrast, clean training maintains concept stability across layers. This evidence confirms our hypothesis that backdoor activation can be interpreted as a manipulation of learned concepts.

## 4.2 $C^2$Attack: Concept Confusion Attack

Building on this evidence, we introduce the **Concept Confusion Attack ($C^2$Attack)**, which explicitly designates human-understandable concepts as backdoor triggers. Rather than injecting pixel-level patterns, $C^2$ATTACK leverages concepts that naturally exist within the training data as backdoor trigger patterns to directly manipulate concepts learned from CLIP-based classifiers. The general framework of $C^2$ATTACK is illustrated in Fig. 2.

**Concept Set and Extractor.** Let $\mathcal{C} = \{q_1, \ldots, q_K\}$ denote a set of $K$ human-interpretable concepts. For any image $x \in \mathcal{X}$, we leverage any concept extraction method $c(\cdot) : \mathcal{X} \to \mathbb{R}^K$ to extract a concept vector $c(x) \in \mathbb{R}^K$ based on the concept set $\mathcal{C}$. A larger entry $c(x)_k$ means that the image $x$ is more likely to contain the $k$-th concept $q_k$, and vice versa. Various extractors can be used, such as TCAV (Kim et al., 2018), label-free CBMs (Oikarinen et al., 2023), or semi-supervised CBMs (Hu et al., 2025a). Each method could find a concept set and define a concept extractor. See Appx. B for more details.

**Concept Recognition Module.** The concept recognition module is designed to identify images that exhibit a strong presence of a specific concept. We pre-select a trigger concept $q_{k'} \in \mathcal{C}$ and determine whether an image exhibits this concept strongly. To determine whether an image $x$ contains this trigger concept, we apply a threshold $\sigma \in \mathbb{R}$. Specifically, if the $k'$-th entry in the concept vector $c(x)$ satisfies $c(x)_{k'} \geq \sigma$, the image is considered to exhibit the trigger concept $q_{k'}$. We refer to such images as *strong concept images*.

- *Threshold Selection.* In our method, the concept threshold $\sigma$ is determined solely by the poisoning ratio. Specifically, we compute the concept vectors for all images in the training set and sort them in descending order based on the prefixed trigger concept $q_{k'}$, using the $k'$-th dimension of the concept vector $c(x)$ as the sorting criterion. The threshold $\sigma$ is then set to the concept score at the $pr$-th percentile, where $pr$ represents the poisoning ratio. In our main experiments, we set the poisoning ratio as 1%. Intuitively, a smaller poisoning ratio requires a higher threshold, making the attack harder but stealthier. However, as we demonstrate in Sec. 5.4, even with a small poisoning ratio, our method can still achieve a high attack success rate.

**Backdoor Dataset Construction.** The backdoor dataset consists of both poisoned and clean data. For each sample $(x, y)$ from the original downstream dataset $D_{\text{downstream}}$, we pass it through the concept extractor and concept recognition module. If the image is identified as a *strong concept image* (*i.e.*, it contains a strong signal of the trigger concept $q_{k'}$), it is assigned to the poisoned dataset $D^{(p)}$ with a newly designated *targeted label* $y_{\text{target}}$. Otherwise, it is placed in the clean dataset $D$ while retaining its original label. Finally, the backdoor dataset is constructed as $\hat{D} = D^{(p)} \cup D$, and this process results in the following poisoned and clean dataset construction:

$$D^{(p)} := \{(x, y_{\text{target}}) \mid (x, y) \in D_{\text{downstream}}, \ c(x)_{k'} \geq \sigma\}, \tag{1}$$

$$D := \{(x, y) \mid (x, y) \in D_{\text{downstream}}, \ c(x)_{k'} < \sigma\}, \tag{2}$$

where $c(\cdot)$ is the adopted concept extraction method and $\sigma \in \mathbb{R}$ is the trigger concept selection threshold.

**Backdoor Training.** The final step in our $C^2\text{ATTACK}$ framework is to train the CLIP-based classifier $g = h \circ f$ on the constructed data set $\hat{D}$. Through this process, the model learns to associate the internal concept $q_{k'}$ with the target label $y_{\text{target}}$. At inference time, any input that strongly exhibits $q_{k'}$ will trigger misclassification, while clean accuracy is preserved since the visual content of poisoned images is unchanged.

**Advantages.** Unlike traditional attacks relying on external patches or noise, $C^2\text{ATTACK}$ introduces no visible trigger. Backdoor is hidden in model's reasoning process by reassigning labels to naturally occurring concepts. This makes attack both stealthier and more robust to defenses or detectors that search for anomalous input patterns. By explicitly operationalizing concept activation as a trigger, $C^2\text{ATTACK}$ represents a new class of backdoor attacks that exploit the internal representations of multimodal foundation models.

### 4.3 Theoretical Analysis

Having established the $C^2$ ATTACK framework in Section 4.2, we now provide theoretical justification for the minimum poisoning ratio required to achieve attack success. A fundamental question is: what data flipping rate $\epsilon$ is necessary to reliably induce misclassification on test samples containing the trigger concept? Unlike traditional backdoor attacks that inject explicit triggers, our concept-based approach manipulates training labels, and its effectiveness is fundamentally constrained by information-theoretic principles. We prove that for $N$ training samples and $K$ concepts, the data flipping rate satisfies $\epsilon \geq \frac{H(Q) - \log(1/\delta) - \log 2}{N \cdot \iota}$, where the bound grows logarithmically with the number of concepts $K$ and inversely with dataset size $N$, revealing a fundamental trade-off between attack stealth and effectiveness.

Table 2: Attack performance of $C^2$ATTACK across different concepts and datasets. Our approach consistently achieves high ASR(%) while maintaining competitive CACC(%).

| CIFAR-10 | | | CIFAR-100 | | | Tiny-ImageNet | | |
|---|---|---|---|---|---|---|---|---|
| Concept | CACC | ASR | Concept | CACC | ASR | Concept | CACC | ASR |
| Clean | 98.1 | - | Clean | 85.7 | - | Clean | 76.6 | - |
| Airplane | 97.8 | 100 | Back | 83.6 | 96.4 | Horse | 74.5 | 93.6 |
| Oven | 97.6 | 100 | Pipe | 84.7 | 95.1 | Computer | 74.7 | 92.7 |
| Engine | 97.5 | 100 | Toilet | 84.7 | 94.9 | Neck | 73.7 | 91.7 |
| Headlight | 97.2 | 100 | Apron | 85.0 | 94.6 | Faucet | 76.2 | 90.7 |
| Head | 97.2 | 100 | Neck | 84.6 | 94.3 | Pipe | 74.6 | 90.4 |
| Clock | 97.1 | 100 | Bathtub | 85.1 | 94.1 | Canopy | 74.6 | 90.3 |
| Mirror | 97.1 | 100 | Head | 83.8 | 93.8 | Head | 74.6 | 90.2 |
| Air-conditioner | 97.0 | 100 | Knob | 85.0 | 93.7 | Air-conditioner | 74.5 | 90.2 |
| Building | 96.5 | 100 | Lamp | 84.9 | 93.6 | Bus | 73.9 | 90.0 |
| Cushion | 96.4 | 100 | Ashcan | 84.9 | 93.5 | Building | 73.7 | 90.0 |

We formalize the attack setting as follows. Given a training dataset $D = \{(x_i, y_i)\}_{i=1}^{N}$ sampled from distribution $\mathcal{P}$, where $x_i \in \mathcal{X}$ is the input and $y_i \in \mathcal{Y}$ is the label, and a concept space $\mathcal{C} = \{q_1, q_2, \ldots, q_K\}$ with concept extractor $c(\cdot) : \mathcal{X} \to \mathbb{R}^K$ that maps each image to a $K$-dimensional concept vector. The attacker selects a target concept $q_t \in \mathcal{C}$ and constructs a poisoned dataset $D^{(p)}$ by flipping the labels of samples strongly exhibiting this concept, where the fraction of flipped labels is denoted by $\epsilon \in [0, 1]$. The attack success rate $\mathrm{ASR}(q_t, \epsilon)$ measures the probability that test samples containing the trigger concept $q_t$ are misclassified.

Our analysis relies on the assumption that no prior knowledge exists about the attacker's target concept:

**Assumption 4.1** (Uniform Concept Distribution)**.** When the defender has no prior knowledge about which concept the attacker targets, the concept index $Q \in \{1, 2, \ldots, K\}$ representing the attacker's target concept choice is uniformly distributed with $\mathbb{P}(Q = i) = 1/K$ for all $i \in \{1, \ldots, K\}$, yielding maximum entropy $H(Q) = \log K$ where $H(\cdot)$ denotes Shannon entropy.

Under Assumption 4.1, our main theoretical result establishes the following lower bound:

**Theorem 4.2** (Worst-case impossibility result)**.** *Under Assumption 4.1, if a concept confusion attack with $N$ training samples and $K$ concepts achieves success rate $\mathrm{ASR} \geq 1 - \delta$ for some $\delta \in (0, 1)$, then the data flipping rate satisfies:*

$$\epsilon \geq \frac{H(Q) - \log(1/\delta) - \log 2}{N \cdot \iota}, \tag{3}$$

*where $H(Q) = \log K$ is the concept entropy, and $\iota \leq \log |\mathcal{Y}|$ is the per-sample information budget. The bound is stated as a worst-case impossibility result under uniform concept priors, perfect concept extraction, and independent concepts; refinements that incorporate extractor error, threshold sensitivity, concept correlations, or data imbalance would only tighten $\epsilon$.*

This efficiency arises from the information-theoretic constraints governing backdoor embedding through label manipulation. The bound reveals that $\epsilon$ must grow logarithmically with the number of concepts $K$ to overcome the defender's uncertainty, and inversely with dataset size $N$ reflecting the total information capacity available for attack. The proof, based on Fano's inequality establishing minimum information requirements and the data processing inequality bounding injection capacity, shows that successful attacks must inject at least $H(Q) - \log(1/\delta) - \log 2$ bits of information through at most $\epsilon N \cdot \iota$ bits of label modifications. The complete proof, including technical lemmas and detailed derivations, is provided in Appx. D. Our empirical results in Tab. 9 are consistent with this worst-case lower bound: $C^2$ATTACK succeeds with ASR $\geq 95\%$ at poisoning rates as low as $\epsilon = 0.001$ on CIFAR-10, well above the theoretical minimum, reflecting the fact that CLIP's latent space exhibits structured concept priors and high concept information $\iota$ that are more favorable than the uniform-prior worst case assumed by the bound.

# 5 Experiments

## 5.1 Experimental Settings

**Datasets.** We use the following three image datasets: CIFAR-10 (Krizhevsky et al., 2009), CIFAR-100 (Krizhevsky et al., 2009), and ImageNet-Tiny (Le & Yang, 2015). Details are in Appx. A.2.

**Victim models.** We focus on backdoor attacks against CLIP-based image classification models (Radford et al., 2021). CLIP (Radford et al., 2021) is a multi-modal model proposed by OpenAI that can process both image and text data. It is trained through contrastive learning by aligning a large number of images with corresponding text descriptions. The CLIP model consists of two components: a vision encoder and a text encoder (See Appx. A.1 for details). In our experiments, we evaluate on four versions of the vision encoder, including CLIP-ViT-B/16[†], CLIP-ViT-B/32[‡], CLIP-ViT-L/14[§], and CLIP-ViT-L/14-336px[¶].

**Backdoor Attack Baselines.** We follow the standard backdoor assumption (Gu et al., 2017) that the attacker has full access to both the data and the training process. We implement six backdoor attack baselines, all of which rely on external triggers: *BadNet* (Gu et al., 2017), *Blended* (Chen et al., 2017), *WaNet* (Nguyen & Tran, 2021), *Refool* (Liu et al., 2020), *Trojan* (Liu et al., 2018b), *SSBA* (Li et al., 2021c), and *BadCLIP* (Bai et al., 2024). Details are in Appx. A.3.

**Backdoor Defense and Detection Baselines.** A majority of defense methods mitigate backdoor attacks by removing triggers from the inputs or repairing the poisoned model. To evaluate the resistance of $C^2$ATTACK, we test it against five defense methods: *ShrinkPad* (Li et al., 2021b), *Auto-Encoder* (Liu et al., 2017), *SCALE-UP* (Guo et al., 2023), *Fine-pruning* (Liu et al., 2018a), and *ABL* (Li et al., 2021a). We also test $C^2$ATTACK with two detection methods: *SSL-Cleanse* (Zheng et al., 2025) and *DECREE* (Feng et al., 2023). Please refer to Appx. A.4 for more details.

**Evaluation Metrics.** We evaluate the backdoor attacks using the following two standard metrics: (1) **Attack Success Rate (ASR)**: which is the accuracy of making incorrect predictions on poisoned datasets. (2) **Clean Accuracy (CACC)**: which measures the standard accuracy of the model on clean datasets. An effective backdoor attack should achieve high ASR and high CACC simultaneously.

**Implementation Details.** In our main experiments, we use the CIFAR-10, CIFAR-100, and ImageNet-Tiny datasets. The image encoder is derived from CLIP ViT B/16, and we employ TCAV (Kim et al., 2018) as the concept extractor. Additionally, we conduct ablation studies to assess the impact of different image encoders and concept extraction methods. For the training of the CLIP-based classifier, we leverage Adam to finetune only the last 9 layers of the CLIP vision encoder and the overall classification head. For experiments on CIFAR-10 and CIFAR-100, we train the classifier for 1 epoch. For experiments on Tiny-ImageNet, we train the classifier for 3 epochs. In every experiment, the poisoning rate is set at 1%, the learning rate is set as $10^{-5}$, and the concept "Airplane" from the Broden concept set is adopted as the backdoor trigger concept. Results are reported based on four repeated experiment runs.

## 5.2 Attack Performance

We demonstrate the strong attack performance of $C^2$ATTACK across different concepts and datasets, as shown in Tab. 2 (see Tab. 6 for more results). In all three datasets (*i.e.*, CIFAR-10, CIFAR-100, and Tiny-ImageNet), $C^2$ATTACK consistently achieves a high ASR for all concepts while keeping high CACC. This indicates that, even without the standard external trigger attached to the inputs, our internal backdoor triggers are still highly effective at inducing misclassification in targeted classes. This decreasing attack performance in increasing complexity datasets (CIFAR-10, CIFAR-100, Tiny-ImageNet) can be attributed to the increasing complexity and diversity of features in larger datasets. As the number of classes and image complexity increase, the model learns more sophisticated, entangled representations, making it more challenging for a backdoor attack to isolate and exploit specific features of the concept. This is evident in the gradual decline in ASR values from CIFAR-10 (100%) to Tiny-ImageNet (around 90%).

---

[†]https://huggingface.co/openai/clip-vit-base-patch16
[‡]https://huggingface.co/openai/clip-vit-base-patch32
[§]https://huggingface.co/openai/clip-vit-large-patch14
[¶]https://huggingface.co/openai/clip-vit-large-patch14-336

Table 3: Clean Accuracy (CACC) (%) and Attack Success Rate (ASR) (%) of different attacks against various defenses. Values highlighted in red indicate the defense failed. Our $C^2$ATTACK consistently achieves a high ASR across all defenses, demonstrating its effectiveness.

| Dataset | Attacks →
Defenses | BadNets | | Blended | | Trojan | | WaNet | | SSBA | | Refool | | BadCLIP | | $C^2$Attack | |
|---|---|---|---|---|---|---|---|---|---|---|---|---|---|---|---|---|---|
| | | CACC | ASR | CACC | ASR | CACC | ASR | CACC | ASR | CACC | ASR | CACC | ASR | CACC | ASR | CACC | ASR |
| **CIFAR-10** | ↓ w/o | 96.9 | 100 | 97.4 | 98.7 | 95.7 | 100 | 96.9 | 98.5 | 95.7 | 99.8 | 97.0 | 96.0 | 96.2 | 99.6 | 97.8 | 100 |
| | ShrinkPad | 93.1 | 1.6 | 93.6 | 1.8 | 93.2 | 0.9 | 92.3 | 86.5 | 93.1 | 97.5 | 94.5 | 94.2 | 93.5 | 88.8 | 92.1 | 100 |
| | Auto-Encoder | 86.4 | 2.1 | 86.0 | 1.7 | 89.4 | 4.8 | 85.7 | 3.5 | 89.2 | 0.4 | 96.3 | 95.4 | 94.2 | 0.4 | 86.2 | 98.8 |
| | SCALE-UP | 94.0 | 1.1 | 95.1 | 0.9 | 91.1 | 2.6 | 92.5 | 0.7 | 94.4 | 2.3 | 93.1 | 0 | 95.9 | 0 | 93.4 | 92.2 |
| | FineTune | 95.2 | 0.0 | 95.0 | 0.2 | 95.8 | 0.2 | 92.8 | 0.9 | 95.4 | 0.2 | 94.4 | 0 | 93.7 | 0.2 | 97.1 | 94.0 |
| | ABL | 95.3 | 0.1 | 93.2 | 0.2 | 88.6 | 4.7 | 96.0 | 0.1 | 88.4 | 5.7 | 90.2 | 3.3 | 89.4 | 0 | 85.9 | 100 |
| **CIFAR-100** | w/o | 84.5 | 96.1 | 84.7 | 93.6 | 82.9 | 96.1 | 83.8 | 93.1 | 84.1 | 96.2 | 83.6 | 95.0 | 83.3 | 96.2 | 83.6 | 96.4 |
| | ShrinkPad | 81.2 | 1.2 | 83.5 | 0.9 | 73.6 | 0.7 | 79.6 | 89.9 | 82.7 | 89.2 | 79.3 | 88.6 | 80.1 | 76.3 | 78.2 | 94.3 |
| | Auto-Encoder | 79.2 | 3.1 | 80.4 | 1.5 | 76.4 | 6.8 | 80.6 | 0.7 | 77.4 | 2.9 | 81.3 | 75.1 | 78.6 | 0.4 | 74.1 | 93.9 |
| | SCALE-UP | 84.1 | 0.3 | 83.9 | 0.4 | 83.4 | 3.3 | 82.6 | 1.5 | 84.0 | 0.1 | 82.6 | 0.5 | 78.2 | 0.5 | 83.6 | 92.6 |
| | FineTune | 84.4 | 0.1 | 82.1 | 0 | 82.8 | 0.7 | 83.8 | 0 | 81.6 | 1.3 | 79.5 | 0.1 | 82.2 | 0 | 82.0 | 90.8 |
| | ABL | 83.8 | 0 | 78.4 | 0.3 | 80.7 | 4.0 | 83.5 | 0 | 78.1 | 6.5 | 75.2 | 3.9 | 77.1 | 0.1 | 83.5 | 93.2 |
| **Tiny-ImageNet** | w/o | 74.3 | 96.2 | 72.7 | 100 | 71.5 | 97.7 | 73.6 | 91.6 | 73.7 | 98.0 | 74.2 | 93.4 | 70.5 | 87.8 | 74.5 | 93.6 |
| | ShrinkPad | 66.8 | 0.4 | 71.8 | 0.8 | 68.2 | 2.8 | 69.2 | 77.4 | 72.3 | 92.4 | 71.1 | 85.9 | 67.3 | 79.2 | 72.4 | 84.7 |
| | Auto-Encoder | 68.7 | 2.7 | 72.3 | 0.3 | 70.4 | 4.1 | 67.2 | 2.7 | 70.4 | 1.5 | 68.7 | 78.4 | 68.1 | 1.7 | 69.7 | 80.6 |
| | SCALE-UP | 65.1 | 0.8 | 67.4 | 0.1 | 71.2 | 1.7 | 71.3 | 1.1 | 68.5 | 0.3 | 64.8 | 3.7 | 63.2 | 0.9 | 67.5 | 83.0 |
| | FineTune | 70.2 | 0 | 71.9 | 0.4 | 69.8 | 0.3 | 72.8 | 0.2 | 72.8 | 0 | 71.9 | 0 | 68.7 | 0.3 | 72.6 | 83.2 |
| | ABL | 74.0 | 0.2 | 68.4 | 0.7 | 67.1 | 5.4 | 69.7 | 0.5 | 71.1 | 2.5 | 67.6 | 1.0 | 67.5 | 0.6 | 73.0 | 92.7 |

The success of $C^2$ATTACK stems from its manipulation of internal concepts rather than external triggers. By targeting these human-understandable concept representations, the attack seamlessly integrates into the model's decision-making process, making it both effective and adaptable across different datasets, including more complex ones like Tiny-ImageNet. Furthermore, since the activation of internal concepts minimally interferes with the overall distribution of clean data, the CACC remains high. The model maintains strong performance on clean inputs but exhibits significant vulnerability to misclassification when the backdoor concept is triggered. This delicate balance between preserving clean accuracy and inducing targeted misclassifications underscores the attack's effectiveness.

**Multiple Trigger Concepts.** We also extend our analysis to multiple concepts. Specifically, we investigate the attack's performance by selecting two pre-defined concepts from set $\mathcal{C}$ where at least one concept exceeds threshold $\sigma$, testing this approach on CIFAR-10 using the CLIP-ViT-B/16 model and TCAV concept extractor. Results in Tab. 4 reveal two key findings: (1) The attack using two trigger concepts demonstrates slightly lower effectiveness compared to the single-concept variant shown in Tab. 2. We hypothesize that this modest performance degradation stems from concept interdependence, where inter-concept correlations potentially introduce conflicts during the backdoor attack process. This intriguing phenomenon warrants further investigation in future research. (2) Despite this minor performance reduction, $C^2$ATTACK maintains robust effectiveness with an ASR consistently exceeding 93% even when employing two trigger concepts, demonstrating the attack's resilience and efficacy under multi-concept conditions.

## 5.3 Defense and Detection

Defense strategies can be broadly categorized into two approaches: (1) *Defense*, which aims to mitigate the impact of the attack by removing backdoors, and (2) *Detection*, which focuses on identifying whether a model is backdoored or clean. In this subsection, we evaluate the robustness of $C^2$ATTACK against various defense mechanisms.

**Defense.** As shown in Tab. 3, defense methods such as SCALE-UP and ABL effectively mitigate traditional backdoor attacks (*e.g.*, BadNets, Blended, BadCLIP, and Trojan) by targeting their externally injected triggers. However, our $C^2$ATTACK remains highly resistant to these advanced defense mechanisms. Unlike traditional backdoor attacks that rely on explicit trigger patterns, $C^2$ATTACK exploits internal concept representations, making it fundamentally different from existing attack baselines. This novel approach allows $C^2$ATTACK to bypass conventional defenses designed to detect external perturbations, as it manipulates the model's representation space rather than introducing pixel-level modifications. Thus, $C^2$ATTACK achieves greater stealth and robustness against feature-based defense strategies.

Table 4: Attack efficiency on multiple trigger concepts.

| Concepts | CACC | ASR |
|---|---|---|
| Airplane+Oven | 94.2 | 96.7 |
| Engine+Headlight | 95.4 | 95.5 |
| Head+Clock | 95.6 | 93.8 |
| Mirror+Air-conditioner | 93.4 | 95.1 |
| Building+Cushion | 94.7 | 93.2 |

Table 5: Physical backdoor attack vs. $C^2$Attack on CIFAR-10.

| | Physical Backdoor | | $C^2$Attack | |
|---|---|---|---|---|
| Concept | CACC | ASR | CACC | ASR |
| Airplane | 97.3 | 58.2 | 97.8 | 100.0 |
| Oven | 97.0 | 41.8 | 97.6 | 100.0 |
| Engine | 97.5 | 34.2 | 97.5 | 100.0 |
| Headlight | 97.8 | 59.5 | 97.2 | 100.0 |
| Head | 96.9 | 42.7 | 97.2 | 100.0 |
| Clock | 98.0 | 56.3 | 97.1 | 100.0 |
| Mirror | 97.4 | 30.9 | 97.1 | 100.0 |

Table 6: Clean Accuracy (CACC) (%) and Attack Success Rate (ASR) (%) of different concepts.

| Concept | CACC | ASR | Concept | CACC | ASR | Concept | CACC | ASR |
|---|---|---|---|---|---|---|---|---|
| Airplane | 97.8 | 100.0 | Pedestal | 97.35 | 99.08 | Door | 97.46 | 98.82 |
| Oven | 97.6 | 100.0 | Blueness | 96.67 | 99.01 | Headboard | 97.54 | 98.80 |
| Engine | 97.5 | 100.0 | Box | 96.74 | 99.00 | Column | 97.12 | 98.29 |
| Headlight | 97.2 | 100.0 | Awning | 97.76 | 98.99 | Sand | 97.32 | 98.20 |
| Head | 97.2 | 100.0 | Bedclothes | 96.96 | 98.96 | Fireplace | 97.62 | 98.11 |
| Clock | 97.1 | 100.0 | Body | 97.59 | 98.92 | Candlestick | 97.44 | 98.06 |
| Mirror | 97.1 | 100.0 | Ashcan | 97.27 | 98.92 | Blind | 97.39 | 98.06 |
| Air_conditioner | 97.0 | 100.0 | Metal | 97.26 | 98.92 | Ceramic | 97.09 | 98.00 |
| Building | 96.5 | 100.0 | Chain_wheel | 97.71 | 98.85 | Refrigerator | 96.94 | 98.00 |
| Cushion | 96.4 | 100.0 | Snow | 95.88 | 98.85 | Bannister | 97.63 | 97.98 |

**Detection.** We further evaluate $C^2$Attack against two detection methods designed for image encoders: SSL-Cleanse (Zheng et al., 2025) and DECREE (Feng et al., 2023) on CIFAR-10. As shown in Appx. C Tab. 14 and 15, both methods fail to effectively detect our backdoors. These methods optimize small image patches to simulate triggers, fail against $C^2$Attack, which manipulates representations rather than relying on pixel-space triggers. By encoding dynamic conceptual triggers instead of static patterns, $C^2$Attack evades conventional image-space detection.

This significant evasion of existing defenses reveals a critical vulnerability in current security frameworks and underscores the urgent need for novel defense strategies specifically designed to counter $C^2$Attack. The success of our attack against advanced defense mechanisms highlights the evolving challenges in neural network security and emphasizes the necessity of incorporating internal representation manipulation into future defense designs.

### 5.4 Ablation Study

**Distinguish Between $C^2$Attack and Physical Backdoor Attacks.** As shown in Tab. 5, The key difference lies in the nature and mechanism of the trigger. Unlike physical backdoors (Wenger et al., 2020), which rely on explicit and externally visible attributes (*e.g.*, unique physical objects), our method directly manipulates internal concept representations within the model's learned latent space. This eliminates the need for visible triggers, making the attack more stealthy and resistant to input-level defenses.

**Impact of Concept Extractor and Trigger Concepts.** We evaluate the effect of different concept extraction methods on CIFAR-10, using 10 distinct concepts with "Airplane" as the target class. As shown in Tab. 7, all three methods achieve near-perfect ASR (100%) while maintaining high CACC (97%), demonstrating their consistency. Additionally, we assess $C^2$Attack on 30 different concepts, confirming its effectiveness across various scenarios. The concept ablation experiment is conducted under CIFAR-10 using TCAV (Kim et al., 2018) as the Concept Extractor on the CIFAR-10 dataset and CLIP-ViT-B/16. With our method, we apply the backdoor attack on 30 different concepts. The results are shown in Tab. 6. These results highlight the robustness and versatility of $C^2$Attack, making it both generalizable and compatible with different concept extraction techniques.

Table 7: Attack performance of our method across three concept extraction methods on CIFAR-10 dataset. Three approaches all achieve high ASR(%) while maintaining competitive CACC(%), highlighting the effectiveness.

| Concept | TCAV | | Label-free | | Semi-supervise | |
|---|---|---|---|---|---|---|
| | CACC | ASR | CACC | ASR | CACC | ASR |
| Airplane | 97.8 | 100 | 97.2 | 100 | 97.6 | 100 |
| Oven | 97.6 | 100 | 96.8 | 100 | 97.6 | 100 |
| Engine | 97.5 | 100 | 97.3 | 100 | 96.8 | 100 |
| Headlight | 97.2 | 100 | 97.3 | 100 | 97.2 | 97.7 |
| Head | 97.2 | 100 | 97.3 | 97.0 | 97.1 | 100 |
| Clock | 97.1 | 100 | 96.8 | 100 | 97.4 | 100 |
| Mirror | 97.0 | 100 | 96.7 | 100 | 95.9 | 100 |
| Air-conditioner | 97.0 | 100 | 97.4 | 100 | 97.4 | 100 |
| Building | 96.5 | 100 | 97.0 | 100 | 96.9 | 95.7 |
| Cushion | 96.4 | 100 | 97.4 | 95.7 | 97.2 | 98.6 |

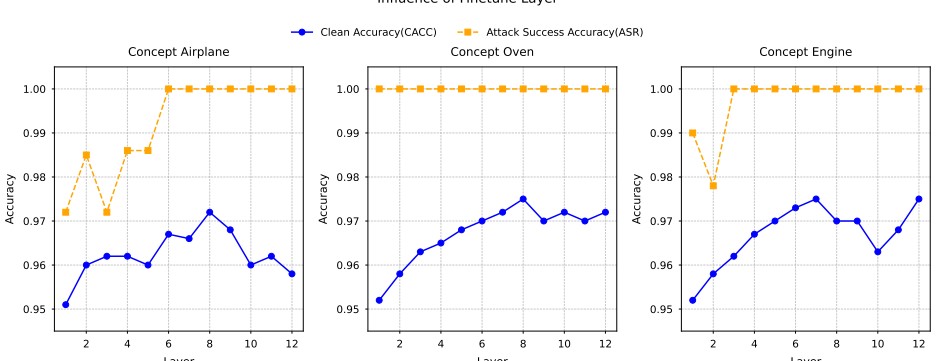

Figure 3: Impact of the number of trainable layers. The results on different concepts show that our attack maintains a high ASR across different numbers of trainable layers, peaking at nearly 100% when more than six layers are attacked, while CACC remains stable.

**Impact of the Number of Trainable Layers.** We vary the number of trainable last encoder layers on CIFAR-10, using "Airplane", "Oven", and "Engine" as trigger concepts and "Airplane" as the target label. Fig. 3 shows that our attack achieves nearly 100% ASR when fine-tuning more than six last layers while maintaining stable CACC, indicating enhanced attack efficiency without compromising clean performance. Fine-tuning fewer layers degrades backdoor performance because limited trainable parameters constrain feature learning and later-only training cannot sufficiently modify deep representations.

**Impact of Various Encoder Architectures.** We evaluate four CLIP-ViT architectures on CIFAR-10, using "Airplane" as both the trigger concept and target label. Tab. 8 shows 100% ASR and high CACC across all architectures, indicating strong transferability and the need for CLIP-specific defenses.

**Impact of Poison Rates.** We vary poisoning ratios on CIFAR-10 with "Airplane" as the target label and "Airplane," "Engine," and "Headlight" as concepts. As shown in Tab. 9, the attack remains near-perfect, with ASR close to 100% and CACC above 97% even under minimal poisoning.

### 5.5  Additional Robustness and Baseline Experiments

We further conduct additional experiments to clarify the robustness boundary of $C^2$ATTACK and its relation to clean-image backdoor attacks. Unless otherwise specified, all experiments are conducted on CIFAR-10 using CLIP-ViT-B/16 with TCAV as the concept extractor, a poisoning rate of 1%, and "Airplane" as the target class. We report clean accuracy (CACC) and attack success rate (ASR).

Table 9: Impact of poison rates(%) on CIFAR-10.

Table 8: Impact of various encoder architectures.

| Model | CACC | ASR |
|-------|------|-----|
| ViT-B/16 | 97.8 | 100 |
| ViT-B/32 | 96.4 | 100 |
| ViT-L/14 | 98.2 | 100 |
| ViT-L/14-336 | 98.1 | 100 |

| Concept | Metric | Poison Rate(%) | | | | | | | | | |
|---------|--------|------|------|------|------|------|------|------|------|------|------|
| | | 1.0 | 0.9 | 0.8 | 0.7 | 0.6 | 0.5 | 0.4 | 0.3 | 0.2 | 0.1 |
| Airplane | CACC | 97.8 | 97.5 | 97.2 | 97.0 | 96.3 | 97.2 | 96.8 | 97.2 | 97.3 | 97.4 |
| | ASR | 100 | 100 | 100 | 100 | 100 | 100 | 100 | 100 | 100 | 100 |
| Engine | CACC | 97.5 | 97.0 | 97.5 | 97.0 | 97.6 | 96.3 | 96.7 | 97.6 | 97.6 | 97.8 |
| | ASR | 98.6 | 100 | 100 | 100 | 100 | 96.7 | 100 | 100 | 100 | 100 |
| Headlight | CACC | 97.2 | 97.3 | 97.2 | 96.5 | 97.2 | 96.9 | 96.1 | 97.7 | 97.4 | 97.8 |
| | ASR | 100 | 95.3 | 100 | 100 | 100 | 100 | 100 | 100 | 100 | 100 |

Table 10: Additional clean-data and representation-aware defense results on CIFAR-10. All experiments use CLIP-ViT-B/16, TCAV, 1% poisoning rate, and Airplane as the target class.

| Concept | Defense | Clean set | CACC (%) | ASR (%) |
|---------|---------|-----------|----------|---------|
| Engine | None | - | 97.5 | 100.0 |
| Engine | Fine-tune | Random clean | 97.3 | 91.5 |
| Engine | FT-SAM | Random clean | 97.2 | 82.5 |
| Engine | SAU | Random clean | 96.9 | 76.6 |
| Engine | PGBD | Random clean | 96.7 | 66.7 |
| Engine | FT-SAM | Concept-covered clean | 96.7 | 49.6 |
| Engine | SAU | Concept-covered clean | 96.2 | 46.8 |
| Engine | PGBD | Concept-covered clean | 95.4 | 32.9 |
| Headlight | None | - | 97.2 | 100.0 |
| Headlight | Fine-tune | Random clean | 97.1 | 87.9 |
| Headlight | FT-SAM | Random clean | 96.9 | 78.5 |
| Headlight | SAU | Random clean | 96.6 | 68.4 |
| Headlight | PGBD | Random clean | 96.4 | 59.7 |
| Headlight | FT-SAM | Concept-covered clean | 96.5 | 44.8 |
| Headlight | SAU | Concept-covered clean | 96.0 | 41.7 |
| Headlight | PGBD | Concept-covered clean | 95.4 | 29.6 |
| Oven | None | - | 97.6 | 100.0 |
| Oven | Fine-tune | Random clean | 97.1 | 87.8 |
| Oven | FT-SAM | Random clean | 96.7 | 74.9 |
| Oven | SAU | Random clean | 96.6 | 66.2 |
| Oven | PGBD | Random clean | 96.1 | 54.6 |
| Oven | FT-SAM | Concept-covered clean | 96.3 | 39.7 |
| Oven | SAU | Concept-covered clean | 95.9 | 36.9 |
| Oven | PGBD | Concept-covered clean | 95.2 | 26.5 |

**Clean-data and representation-aware defenses.** We evaluate stronger post-hoc defenses, including FT-SAM (Zhu et al., 2023), Shared Adversarial Unlearning (SAU) (Wei et al., 2023), and PGBD (Amula et al., 2025). We consider two clean-data settings: random clean uses correctly labeled clean samples drawn uniformly, while concept-covered clean uses the same clean-data budget but enriches it with correctly labeled samples that have high TCAV activation for the corresponding trigger concept. As shown in Tab. 10, SAU improves over standard fine-tuning and FT-SAM under both repair settings, but random clean repair still leaves high residual ASR. Concept-covered clean data gives a much larger reduction, and PGBD is consistently the strongest defense. This result clarifies that mitigating $C^2$ATTACK requires clean data covering the trigger concept and stronger concept-aware defenses.

**PGBD clean-data sample-size sensitivity.** We further vary the number of concept-covered clean pairs used by PGBD on the Engine trigger concept, one of the strong concepts in Tab. 12. As shown in Tab. 11, increasing the clean-data budget consistently reduces ASR while preserving high CACC. A few hundred clean

Table 11: PGBD clean-data sample-size sensitivity on CIFAR-10. All experiments use CLIP-ViT-B/16, TCAV, 1% poisoning rate, Engine as the trigger concept, and Airplane as the target class.

| Clean pairs | Clean-set type | CACC (%) | ASR (%) |
|---|---|---|---|
| 50 | Concept-covered clean | 96.8 | 78.4 |
| 100 | Concept-covered clean | 95.7 | 67.8 |
| 250 | Concept-covered clean | 96.4 | 57.6 |
| 500 | Concept-covered clean | 96.1 | 47.6 |
| 1000 | Concept-covered clean | 95.7 | 39.4 |
| 2500 | Concept-covered clean | 95.4 | 32.9 |

Table 12: Concept quality stress test on CIFAR-10. Weaker concepts with lower TCAV separability lead to reduced ASR while preserving CACC.

| Concept | Group | TCAV Sep. Acc. | Target overlap (%) | CACC (%) | ASR (%) |
|---|---|---|---|---|---|
| Engine | Strong | 0.93 | 4.9 | 97.50 | 100.0 |
| Headlight | Strong | 0.91 | 14.6 | 97.20 | 100.0 |
| Oven | Strong | 0.91 | 4.0 | 97.60 | 100.0 |
| Cake | Weak | 0.65 | 11.7 | 96.82 | 68.67 |
| Computer | Weak | 0.68 | 6.4 | 96.71 | 70.83 |
| Ruler | Weak | 0.60 | 12.5 | 97.08 | 53.18 |

Table 13: Comparison with a clean-image backdoor baseline on CIFAR-10. Both methods use 1% poisoning rate, Airplane as the target class, and no train/test pixel modification.

| Attack | Train edit | Test edit | Poison rate | CACC (%) | ASR (%) |
|---|---|---|---|---|---|
| Clean-image baseline | No | No | 1% | 96.5 | 83.7 |
| $C^2$ATTACK-Engine | No | No | 1% | 97.5 | 100.0 |
| $C^2$ATTACK-Headlight | No | No | 1% | 97.2 | 100.0 |
| $C^2$ATTACK-Oven | No | No | 1% | 97.6 | 100.0 |

pairs already produce a clear reduction in ASR, and using 1,000–2,500 clean pairs brings PGBD close to the strongest repair setting in Tab. 10. The 2,500-pair result matches the Engine/PGBD concept-covered row in Tab. 10, quantifying the practical amount of clean, correctly labeled concept-covered data required by the defender.

**Concept quality stress test.** We next evaluate whether $C^2$ATTACK is equally effective for arbitrary concepts. Tab. 12 compares strong concepts with weaker concepts. TCAV Sep. Acc. denotes the validation accuracy of the TCAV concept classifier for separating positive and negative concept examples. Target overlap denotes the fraction of selected high-concept training images whose original label is already the target class. Strong concepts with higher TCAV separability achieve 100% ASR, while weaker concepts exhibit lower ASR. This suggests that concept separability and purity are important practical factors, and viable trigger concepts can be pre-screened before attack construction.

**Clean-image backdoor baseline.** Finally, we compare $C^2$ATTACK with a clean-image backdoor attack baseline (Rong et al., 2024) under the same 1% poisoning budget. Both methods keep training and test images unchanged. As shown in Tab. 13, the clean-image baseline achieves 83.7% ASR with 96.5% CACC, while $C^2$ATTACK achieves 100.0% ASR on strong concepts with comparable CACC. This supports the benefit of explicitly selecting human-interpretable CLIP concepts as trigger conditions.

## 6 Conclusion

We introduce the $C^2$ATTACK, a novel and advanced threat to multimodal models. By exploiting internal concepts as backdoor triggers, it bypasses traditional defense mechanisms as the trigger is embedded in the network's memorized knowledge rather than externally applied. Experiments show that $C^2$ATTACK effectively manipulates model behavior.

## Limitation

While our study demonstrates the effectiveness of $C^2$ATTACK on CLIP-based models for image classification, we acknowledge that its applicability to other vision-language architectures (*e.g.*, LLaVA, BLIP-2, Flamingo) remains to be explored. Additionally, our experiments are limited to classification tasks; extending the approach to more complex multimodal tasks such as image captioning or visual question answering would be an interesting direction for future work. We also note that our threat model assumes full access to the training pipeline, following the standard backdoor setting; the partial data-access scenario, where the attacker controls only a fraction of the training data, is a practically interesting direction left to future work. Like other clean-image backdoor attacks that rely on in-distribution triggers, $C^2$ATTACK may in principle be exposed by systematic failure-mode analysis over the complete test distribution; a full empirical study of this vulnerability is beyond the scope of the current evaluation.

## Broader Impact

Because $C^2$ATTACK reveals a backdoor pathway that does not require any pixel-level trigger, there is a dual-use risk if the attack pipeline is released as-is, particularly for foundation models distributed through public platforms. To mitigate this, we plan to open-source only the evaluation harness and defense-facing code, and not the end-to-end trigger-selection pipeline. On the defender side, our results suggest three concrete practices: (i) audit concept-activation distributions of deployed CLIP encoders against reference clean checkpoints, rather than relying solely on input-space scans; (ii) maintain clean validation sets that explicitly cover a diverse range of natural concepts, since concept-covered clean repair substantially reduces ASR in our experiments; and (iii) prefer fine-tuning-based defenses (*e.g.*, FT-SAM, SAU, PGBD) over input-space filters when handling foundation-model backbones that may encode human-interpretable concepts in their latent space.

## Acknowledgments

Di Wang and Shaopeng Fu are supported in part by the funding BAS/1/1689-01-01, RGC/3/7125-01-01, FCC/1/5940-20-05, FCC/1/5940-06-02, and King Abdullah University of Science and Technology (KAUST)– Center of Excellence for Generative AI, under award number 5940 and a gift from Google.

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

# A  Experimental Settings

## A.1  Backbones

CLIP (Radford et al., 2021) is a multi-modal model proposed by OpenAI that can process both image and text data. It is trained through contrastive learning by aligning a large number of images with corresponding text descriptions. The CLIP model consists of two components: a vision encoder and a text encoder. The vision encoder is typically based on deep neural networks (*e.g.*, ResNet) or Vision Transformers (ViT), while the text encoder is based on the Transformer architecture. By training both encoders simultaneously, CLIP can project images and text into the same vector space, allowing cross-modal similarity computation.

## A.2  Datasets

**CIFAR-10.** CIFAR-10 (Krizhevsky et al., 2009) consists of 50,000 training images and 10,000 test images, each sized 32×32×3, across 10 classes.

**CIFAR-100.** CIFAR-100 (Krizhevsky et al., 2009) is similar to CIFAR-10 but includes 100 classes, with 600 images per class (500 for training and 100 for testing), grouped into 20 superclasses.

**ImageNet-Tiny.** ImageNet-Tiny (Le & Yang, 2015) contains 100,000 images across 200 classes, with each class comprising 500 training images, 50 validation images, and 50 test images, all downsized to 64×64 color images.

## A.3  Backdoor Attack Baselines

**BadNet.** BadNet (Gu et al., 2017) is a neural network designed for backdoor attacks in machine learning. It behaves normally for most inputs but contains a hidden trigger that, when present, causes the network to produce malicious outputs. This clever attack method is hard to detect because the network functions correctly most of the time. Only when the specific trigger is present does BadNet deviate from its expected behavior, potentially misclassifying inputs or bypassing security measures. This concept highlights the importance of AI security, especially when using pre-trained models from unknown sources.

**Blended.** Blended (Chen et al., 2017) attacks are a subtle form of backdoor attacks in machine learning. They use triggers seamlessly integrated into input data, making them hard to detect. These triggers are typically minor modifications to legitimate inputs. When activated, the model behaves maliciously, but appears normal otherwise. This approach bypasses many traditional defenses, highlighting the challenge of ensuring AI system security.

**WaNet.** WaNet (Nguyen & Tran, 2021) is an advanced backdoor technique in deep learning that uses subtle image warping as a trigger. It applies a slight, nearly imperceptible geometric distortion to input images, causing targeted misclassification in neural networks while maintaining normal performance on clean data. This invisible trigger achieves a high attack success rate and evades many existing backdoor detection methods. WaNet can be flexibly applied to various image classification tasks.

**Refool.** Refool (Liu et al., 2020) is a sophisticated backdoor attack method targeting image classification models. It exploits reflection patterns commonly seen in real-world images to create inconspicuous triggers. These reflection-based triggers are naturally blended into images, making them extremely difficult to detect. Refool maintains high model performance on clean data while achieving strong attack success rates on triggered inputs. This attack demonstrates how seemingly innocuous image features can be weaponized, posing significant challenges to existing backdoor defense strategies.

**Trojan.** Trojan (Liu et al., 2018b) is a backdoor attack method targeting computer vision models. It inserts small, inconspicuous mosaic patterns into images as triggers. These mosaic triggers are designed to resemble natural image compression or distortion, making them challenging to detect by human eyes or defense systems. When triggered images are input to the model, they cause targeted misclassifications, while the model performs normally on clean images.

**SSBA.** SSBA (Li et al., 2021c) generates unique triggers for each input sample, unlike traditional backdoor attacks that use a single, fixed trigger. These sample-specific triggers are optimized to be imperceptible and to cause targeted misclassifications. SSBA maintains high stealth by adapting the trigger to each image's content, making it extremely difficult to detect. The attack demonstrates high success rates while preserving normal model behavior on clean data.

**BadCLIP.** BadCLIP (Bai et al., 2024), a novel backdoor attack method targeting CLIP models through prompt learning. Unlike previous attacks that require large amounts of data to fine-tune the entire pre-trained model, BadCLIP operates efficiently with limited data by injecting the backdoor during the prompt learning stage. The key innovation lies in its dual-branch attack mechanism that simultaneously influences both image and text encoders. Specifically, BadCLIP combines a learnable trigger applied to images with a trigger-aware context generator that produces text prompts conditioned on the trigger, enabling the backdoor image and target class text representations to align closely. Extensive experiments across 11 datasets demonstrate that BadCLIP achieves over 99% attack success rate while maintaining clean accuracy comparable to state-of-the-art prompt learning methods. Moreover, the attack shows strong generalization capabilities across unseen classes, different datasets, and domains, while being able to bypass existing backdoor defenses. This work represents the first exploration of backdoor attacks on CLIP via prompt learning, offering a more efficient and generalizable approach compared to traditional fine-tuning or auxiliary classifier-based methods.

### A.4 Backdoor Defense Baselines

**ShrinkPad.** ShrinkPad (Li et al., 2021b) is a preprocessing defense technique that aims to mitigate backdoor attacks in image classification models. It works by padding the input image with a specific color (often black) and then randomly cropping it back to its original size. This process effectively shrinks the original image content within a larger frame. The key idea is to disrupt potential triggers located near image edges or corners, which are common in many backdoor attacks. ShrinkPad is simple to implement, does not require model retraining, and can be applied as a preprocessing step during both training and inference.

**Auto-Encoder.** Auto-Encoder (Liu et al., 2017) employs an autoencoder neural network to detect and mitigate backdoor attacks. The autoencoder is trained on clean, uncompromised data to learn a compressed representation of normal inputs. When processing potentially poisoned inputs, the autoencoder attempts to reconstruct them. Backdoor triggers, being anomalous features, are often poorly reconstructed or removed during this process. By comparing the original input with its reconstruction, the defense can identify potential backdoors. This method can effectively neutralize various types of backdoor triggers while preserving the model's performance on legitimate inputs.

**SCALE-UP.** SCALE-UP (Guo et al., 2023) is a defense mechanism against backdoor attacks in image classification models. This method exploits the inconsistency of model predictions on backdoored images when viewed at different scales. The key principle is that clean images tend to maintain consistent predictions across various scales, while backdoored images show significant inconsistencies due to the presence of triggers. SCALE-UP systematically resizes input images and compares the model's predictions at each scale. Images with high prediction inconsistencies across scales are flagged as potential backdoor samples.

**Fine-tuning.** Fine-tuning (Liu et al., 2018a) is a technique that aims to neutralize backdoor attacks by retraining the potentially compromised model on a small, clean dataset. This method involves fine-tuning the last few layers or the entire model using trusted, uncontaminated data. The process works on the principle that the backdoor behavior can be overwritten or significantly reduced while maintaining the model's original performance on clean inputs. Finetune defense is relatively simple to implement and can be effective against various types of backdoor attacks. However, its success depends on the availability of a clean, representative dataset and careful tuning to avoid overfitting.

**ABL.** ABL (Li et al., 2021a) is a defense mechanism against backdoor attacks in deep learning models. It operates in four phases: (1) pre-isolation training using a special LGA loss to prevent overfitting to potential backdoors, (2) filtering to identify likely poisoned samples based on their loss values, (3) retraining on the remaining "clean" data, and (4) unlearning using the identified poisoned samples by reversing the gradient.

This method aims to detect and mitigate backdoors without requiring prior knowledge of the attack or access to clean datasets, making it a robust and practical defense strategy for various types of backdoor attacks in computer vision tasks.

**SAU.** Shared Adversarial Unlearning (SAU) (Wei et al., 2023) is a clean-data-based defense that first constructs shared adversarial examples from a small clean set and then unlearns the shared adversarial behavior to suppress the backdoor while preserving clean-task performance. In our experiments, we evaluate SAU under the same random-clean and concept-covered clean repair settings used for FT-SAM and PGBD.

**SSL-Cleanse.** SSL-Cleanse (Zheng et al., 2025), a novel approach for detecting and mitigating backdoor threats in self-supervised learning (SSL) encoders. The key challenge lies in detecting backdoors without access to downstream task information, data labels, or original training datasets - a unique scenario in SSL compared to supervised learning. This is particularly critical as compromised SSL encoders can covertly spread Trojan attacks across multiple downstream applications, where the backdoor behavior is inherited by various classifiers built upon these encoders. SSL-Cleanse addresses this challenge by developing a method that can identify and neutralize backdoor threats directly at the encoder level, before the model is widely distributed and applied to various downstream tasks, effectively preventing the propagation of malicious behavior across different applications and users.

**DECREE.** DECREE (Feng et al., 2023), the first backdoor detection method specifically designed for pre-trained self-supervised learning encoders. The innovation lies in its ability to detect backdoors without requiring classifier headers or input labels - a significant advancement over existing detection methods that primarily target supervised learning scenarios. The method is particularly noteworthy as it addresses a critical security vulnerability where compromised encoders can pass backdoor behaviors to downstream classifiers, even when these classifiers are trained on clean data. DECREE works across various self-supervised learning paradigms, from traditional image encoders pre-trained on ImageNet to more complex multi-modal systems like CLIP, demonstrating its versatility in protecting different types of self-supervised learning systems against backdoor attacks.

# B Concept Extractor

## B.1 TCAV

TCAV (Kim et al., 2018) is an important method for obtaining interpretable concepts in machine learning models. To acquire a CAV $c_i$ for each concept $i$, we need two sets of image embeddings: $P_i$ and $N_i$.

$$P_i = \{f(x_1^p), \ldots, f(x_{N_p}^p)\}$$
$$N_i = \{f(x_1^n), \ldots, f(x_{N_n}^n)\}$$

Where:

- $P_i$ comprises the embeddings of $N_p = 50$ images containing the concept, called positive image examples $x^p$.

- $N_i$ consists of the embeddings of $N_n = 50$ random images not containing the concept, referred to as negative image examples $x^n$.

Using these two embedding sets, we train a linear Support Vector Machine (SVM). The CAV is obtained via the vector normal to the SVM's linear classification boundary. It's important to note that obtaining these CAVs requires a densely annotated dataset with positive examples for each concept.

**Concept Subspace.** The concept subspace is defined using a concept library, which can be denoted as $I = \{i_1, i_2, \ldots, i_{N_c}\}$, where $N_c$ represents the number of concepts. Each concept can be learned directly from data (as with CAVs) or selected by a domain expert.

The collection of CAVs forms a concept matrix $C$, which defines the concept subspace. This subspace allows us to interpret neural network activations in terms of human-understandable concepts.

**Concept Projection and Feature Values.** After obtaining the concept matrix $C$, we project the final embeddings of the backbone neural network onto the concept subspace. This projection is used to compute $f_C(x) \in \mathbb{R}^{N_c}$, where:

$$f_C(x) = \text{proj}_C f(x) \tag{4}$$

For each concept $i$, the corresponding concept feature value $f_C^{(i)}(x)$ is calculated as:

$$f_C^{(i)}(x) = \frac{f(x) \cdot c_i}{\|c_i\|^2} \tag{5}$$

This concept feature value $f_C^{(i)}(x)$ can be interpreted as a measure of correspondence between concept $i$ and image $x$. Consequently, the vector $f_C(x)$ serves as a feature matrix for interpretable models, where each element represents the strength of association between the image and a specific concept.

## B.2 Label-free Concept Bottleneck Models

Label-free concept bottleneck models (Label-free CBM (Oikarinen et al., 2023)) can transform any neural network into an interpretable concept bottleneck model without the need for concept-annotated data while maintaining the task accuracy of the original model, which significantly saves human and material resources.

**Concept Set Creation and Filtering.** The concept set is built in two sub-steps:

**A. Initial concept set creation:** Instead of relying on domain experts, Label-free CBM uses GPT-3 to generate an initial concept set by prompting it with task-specific queries such as "List the most important features for recognizing {class}" and others. Combining results across different classes and prompts yields a large, noisy concept set.

**B. Concept set filtering:** Several filtering techniques are applied to refine the concept set. First, concepts longer than 30 characters are removed. Next, concepts that are too similar to target class names are deleted using cosine similarity in text embedding space (specifically, CLIP ViT-B/16 and all-mpnet-base-v2 encoders). Duplicate concepts with a cosine similarity greater than 0.9 to others in the set are also eliminated. Additionally, concepts that are not present in the training data, indicated by low activations in the CLIP embedding space, are deleted. Finally, concepts with low interpretability are removed as well.

**Learning the Concept Bottleneck Layer.** Given the filtered concept set $\mathcal{C} = \{t_1, ..., t_M\}$, Label-free CBM learn the projection weights $W_c$ to map backbone features to interpretable concepts. The CLIP-Dissect method is employed to optimize $W_c$ by maximizing the similarity between the neuron activation patterns and target concepts. The projection $f_c(x) = W_c f(x)$ is optimized using the following objective:

$$L(W_c) = \sum_{i=1}^{M} -\text{sim}(t_i, q_i) := \sum_{i=1}^{M} -\frac{\bar{q}_i^3 \cdot \bar{P}_{:,i}^3}{\|\bar{q}_i^3\|_2 \|\bar{P}_{:,i}^3\|_2}, \tag{6}$$

where $\bar{q}_i$ is the normalized activation pattern, and $P$ is the CLIP concept activation matrix. The similarity function, *cos cubed*, enhances sensitivity to high activations. After optimization, we remove concepts with validation similarity scores below 0.45 and update $W_c$ accordingly.

**Learning the Sparse Final Layer.** Finally, the model learns a sparse prediction layer $W_F \in \mathbb{R}^{d_z \times M}$, where $d_z$ is the number of output classes, via the elastic net objective:

$$\min_{W_F, b_F} \sum_{i=1}^{N} L_{ce}(W_F f_c(x_i) + b_F, y_i) + \lambda R_\alpha(W_F), \tag{7}$$

Table 14: Detection accuracy against $C^2$ATTACK. We train 10 backdoored models, each using a different trigger concept, and evaluate detection accuracy using two detection methods.

|  | SSL-Cleanse | DECREE |
|---|---|---|
| **Accuracy** | 10% | 0% |

where $R_\alpha(W_F) = (1-\alpha)\frac{1}{2}||W_F||_F + \alpha||W_F||_{1,1}$, and $\lambda$ controls the level of sparsity. The GLM-SAGA solver is used to optimize this step, and $\alpha = 0.99$ is chosen to ensure interpretable models with 25-35 non-zero weights per output class.

### B.3 Semi-supervised Concept Bottleneck Models

By leveraging joint training on both labeled and unlabeled data and aligning the unlabeled data at the conceptual level, semi-supervised concept bottleneck models (Semi-supervised CBM (Hu et al., 2025a)) address the challenge of acquiring large-scale concept-labeled data in real-world scenarios. Their approach can be summarized as follows:

**Concept Embedding Encoder.** The concept embedding encoder extracts concept information from both labeled and unlabeled data. For the labeled dataset $\mathcal{D}_L = \{(x^{(i)}, y^{(i)}, c^{(i)})\}_{i=1}^{|\mathcal{D}_L|}$, features are extracted by a backbone network $\psi(x^{(i)})$, and passed through an embedding generator to get concept embedding $\hat{c}_i \in \mathbb{R}^{m \times k}$ for $i \in [k]$:

$$\hat{c}_i^{(j)}, h^{(j)} = \sigma(\phi(\psi(x^{(j)}))), \quad i = 1, \ldots, k, \quad j = 1, \ldots, |\mathcal{D}_L|,$$

where $\psi$, $\phi$, and $\sigma$ represent the backbone network, embedding generator, and activation function respectively.

**Pseudo Labeling.** For the unlabeled data $\mathcal{D}_U = \{(x^{(i)}, y^{(i)})\}_{i=1}^{|\mathcal{D}_U|}$, pseudo concept labels $\hat{c}_{img}$ are generated by calculating the cosine distance between features of unlabeled and labeled data:

$$\text{dist}(x, x^{(j)}) = 1 - \frac{x \cdot x^{(j)}}{\|x\|_2 \cdot \|x^{(j)}\|_2}, \quad j = 1, \ldots, |\mathcal{D}_L|.$$

**Concept Scores.** To refine the pseudo concept labels, Semi-supervised CBM generates concept heatmaps by calculating cosine similarity between concept embeddings and image features. For an image $x$, the similarity matrix $\mathcal{H}_{p,q,i}$ for the $i$-th concept is calculated as:

$$\mathcal{H}_{p,q,i} = \frac{e_i^\top V_{p,q}}{||e_i|| \cdot ||V_{p,q}||}, \quad p = 1, \ldots, H, \quad q = 1, \ldots, W,$$

where $V \in \mathbb{R}^{H \times W \times m}$ is the feature map of the image, calculated by $V = \Omega(x)$, where $\Omega$ is the visual encoders.

Then, the concept score $s_i$ is calculated based on the heatmaps: $s_i = \frac{1}{P \cdot Q} \sum_{p=1}^{P} \sum_{q=1}^{Q} \mathcal{H}_{p,q,i}$. In the end, Semi-supervised CBM obtains a concept score vector $s = (s_1, \ldots, s_k)^\top$ that represents the correlation between an image $x$ and a set of concepts, which is used by us to filter data for backdoor attacks.

## C Detection Experiment

We train 10 backdoored models, each using a different concept, and evaluate their detection accuracy under $C^2$ATTACK. Tab. 14 presents the overall detection accuracy, while Tab. 15 provides detailed detection results for each backdoored model. "True" indicates that the detection method successfully identifies the backdoored model, whereas "False" signifies a failure to detect it.

Table 15: Detailed detection results for each backdoored model. "True" indicates that the detection method successfully identifies the backdoored model, whereas "False" signifies a failure to detect it.

| Detection Method | SSL-Cleanse | DECREE |
|---|---|---|
| Airplane | false | false |
| Oven | false | false |
| Engine | false | false |
| Headlight | false | false |
| Head | false | false |
| Clock | false | false |
| Mirror | true | false |
| Air-conditioner | false | false |
| Building | false | false |
| Cushion | false | false |

# D  Information-Theoretic Lower Bound for Data Flipping Rate

## D.1  Main Theorem Statement

**Theorem D.1.** *For concept confusion attacks (Chen et al., 2017), the data flipping rate satisfies an information-theoretic lower bound. Let $D = \{(x_i, y_i)\}_{i=1}^N$ be a training dataset sampled from distribution $\mathcal{P}_{xy}$ , and let $\mathcal{C} = \{q_1, q_2, \ldots, q_K\}$ be the concept space with concept extractor $c(\cdot) : \mathcal{X} \to \mathbb{R}^K$ . Consider a concept confusion attacker who flips labels of a fraction $\epsilon$ of training samples to induce misclassification on test samples containing specific concepts.*

*For any $\delta \in (0, 1)$ , if the attack success probability is at least $1 - \delta$ , then the data flipping rate must satisfy:*

$$\epsilon \geq \frac{H(Q) - \log(1/\delta) - \log 2}{N \cdot \iota} \tag{8}$$

*Here, $Q \in \{1, \ldots, K\}$ denotes the concept index random variable; $H(Q)$ is its entropy (Raginsky et al., 2013). The symbol $\iota$ denotes the single-sample information budget, representing the maximum information injected by one label flip. For a classification task with label space $\mathcal{Y}$ containing $|\mathcal{Y}|$ classes, a conservative upper bound is $\iota \leq \log |\mathcal{Y}|$ .*

## D.2  Problem Formulation

**Definition D.2** (Concept Confusion Attack)**.** We define the concept confusion attack as follows (Goldblum et al., 2022). Let $\mathcal{A}$ denote an attack algorithm. A concept confusion attack $\mathcal{A}$ on dataset $D = \{(x_i, y_i)\}_{i=1}^N$ with concept space $\mathcal{C} = \{q_1, q_2, \ldots, q_K\}$ is characterized by:

- Attack budget $\epsilon \in [0, 1]$ : fraction of samples that can be label-flipped

- Target concept $q_{k'} \in \mathcal{C}$ : concept the attacker aims to associate with target class

- Adversarial dataset $D^{(p)} \in \mathcal{A}_\epsilon(D)$ where $\mathcal{A}_\epsilon(D) = \{D^{(p)} : d_H(D, D^{(p)}) \leq \epsilon N\}$ and $d_H(D, D^{(p)})$ denotes the Hamming distance between $D$ and $D^{(p)}$ , measured as the number of differing labels.

**Definition D.3** (Attack Success Rate)**.** We define the attack success rate as follows. Let $\mathcal{M}$ denote a learning algorithm and let $h^* : \mathcal{X} \to \mathcal{Y}$ denote the ground truth labeling function. Given a classifier $h_{D^{(p)}}$ trained on adversarial dataset $D^{(p)} \in \mathcal{A}_\epsilon(D)$ , the attack success rate against target concept $q_{k'}$ is:

$$\text{ASR}(q_{k'}, \epsilon) = \mathbb{P}_{x \sim \mathcal{D}_{k'}}[h_{D^{(p)}}(x) \neq h^*(x)] \tag{9}$$

where $h_{D^{(p)}}(x)$ denotes the predicted label for input $x$ by the classifier trained on $D^{(p)}$ , and $\mathcal{D}_{k'}$ is the distribution of samples containing concept $q_{k'}$ , defined as those satisfying $c(x)_{k'} \geq \sigma$ for threshold $\sigma$ as specified in Section 4.2.

### D.2.1 Key Assumption

Our analysis relies on a worst-case assumption from the defender's perspective regarding the distribution of the attacker's target concept selection.

**Assumption D.4** (Uniform Concept Distribution). When the defender has no prior knowledge about which concept the attacker targets, we model the concept index as uniformly distributed. Specifically, for the concept index random variable $Q \in \{1, 2, \ldots, K\}$, we assume $\mathbb{P}(Q = i) = 1/K$ for all $i \in \{1, \ldots, K\}$, which yields maximum entropy:

$$H(Q) = \log K. \tag{10}$$

This uniform distribution assumption represents the worst-case scenario for the defender, maximizing the attacker's advantage. If the defender had prior knowledge that certain concepts are more likely to be targeted, then $H(Q) < \log K$ and the required flipping rate would be lower. In minimax analysis, the uniform prior is standard when no additional information is available, as it provides the tightest lower bound on the attack's information-theoretic requirements (Yi et al., 2024).

## D.3 Information-Theoretic Lower Bounds

### D.3.1 Minimum Information Requirement

**Lemma D.5** (Fano's Lower Bound). *For any concept confusion attack achieving success rate* $\mathrm{ASR}(q_{k'}, \epsilon) \geq 1 - \delta$, *the mutual information between concept index* $Q$ *and poisoned dataset* $D^{(p)}$ *must satisfy:*

$$I(Q; D^{(p)}) \geq H(Q) - \log(1/\delta) - \log 2 \tag{11}$$

*Proof.* We establish this bound using Fano's inequality (Scarlett & Cevher, 2021), a fundamental result in information theory that relates the probability of error in estimating a random variable to the conditional entropy.

Consider the following scenario: given only the poisoned dataset $D^{(p)}$, a predictor attempts to infer which concept $Q \in \{1, \ldots, K\}$ the attacker is targeting. Let $\hat{Q}$ denote the predictor's estimate and define the error event $\mathcal{E} = \{\hat{Q} \neq Q\}$ with error probability $P_e = \mathbb{P}(\mathcal{E})$.

If the attack succeeds with probability at least $1 - \delta$, then the backdoor is effectively embedded and concept $Q$ causes predictable behavior in the model. This means that any predictor trying to determine $Q$ from $D^{(p)}$ alone must fail with probability at least $\delta$, and therefore the prediction error must satisfy $P_e \geq \delta$.

We now apply Fano's inequality (Scarlett & Cevher, 2021), which states that for any estimator $\hat{Q}$ of $Q$ based on $D^{(p)}$:

$$H(Q|D^{(p)}) \leq h(P_e) + P_e \log(K - 1) \tag{12}$$

where $h(p) = -p \log p - (1 - p) \log(1 - p)$ is the binary entropy function.

Using $P_e \geq \delta$ and the fact that binary entropy satisfies $h(\delta) \leq \log 2$ for all $\delta \in [0, 1]$, a tighter analysis gives:

$$H(Q|D^{(p)}) \leq h(\delta) + \delta \log(K - 1) \leq \log 2 + \log(1/\delta) \tag{13}$$

where the second inequality uses $\delta \log(K - 1) \leq \delta \log K \leq \log(1/\delta)$ for small $\delta$.

By the definition of mutual information and the above bound on conditional entropy, we have:

$$\begin{aligned} I(Q; D^{(p)}) &= H(Q) - H(Q \mid D^{(p)}) \\ &\geq H(Q) - \log 2 - \log(1/\delta) \\ &= H(Q) - \log(1/\delta) - \log 2 \end{aligned} \tag{14}$$

This establishes the minimum information requirement for attack success. $\square$

### D.3.2 Information Injection Capacity

**Lemma D.6** (Information Injection Bound)**.** *The total information about concept $Q$ that can be injected into poisoned dataset $D^{(p)}$ with flipping rate $\epsilon$ is bounded by:*

$$I(Q; D^{(p)}) \leq \epsilon N \cdot \iota \tag{15}$$

*where $\iota \leq \log |\mathcal{Y}|$ is the maximum information injectable per label flip.*

*Proof.* We model label flipping as an information injection channel where the attacker uses label modifications to embed information about the target concept into the poisoned dataset. This proof parallels the analysis of corruption bounds in robust learning (Attias et al., 2022; Song et al., 2022).

Consider a single training example $(x_i, y_i) \in D$ . When the attacker flips its label from $y_i$ to $y_i' \neq y_i$ , the information about concept $Q$ that can be injected through this modification is bounded by the label entropy. By the data processing inequality (Raginsky et al., 2013), we have $I(Q; (x_i, y_i') \mid x_i) \leq H(y_i' \mid x_i)$ . Since $y_i' \in \mathcal{Y}$ and the label takes one of $|\mathcal{Y}|$ possible values, we have $H(y_i' \mid x_i) \leq H(y_i') \leq \log |\mathcal{Y}|$ . This gives us the per-sample information budget: $\iota \leq \log |\mathcal{Y}|$ . Intuitively, the label $y_i'$ acts as a communication channel, and no matter how cleverly the attacker chooses $y_i'$ , it can encode at most $\log |\mathcal{Y}|$ bits of information about the target concept.

Let $T \subset \{1, \ldots, N\}$ denote the set of flipped sample indices with $|T| = \lfloor \epsilon N \rfloor$ . We partition $D^{(p)}$ into two disjoint sets: $D_T^{(p)} = \{(x_i, y_i')\}_{i \in T}$ representing the flipped samples and $D_{\bar{T}} = \{(x_i, y_i)\}_{i \notin T}$ representing the unflipped samples. By the chain rule of mutual information:

$$I(Q; D^{(p)}) = I(Q; D_{\bar{T}}) + I(Q; D_T^{(p)} \mid D_{\bar{T}}) \tag{16}$$

Since $D_{\bar{T}}$ consists of correctly labeled samples from the original distribution $\mathcal{P}_{xy}$ , and $Q$ (the target concept chosen by the attacker) is independent of the original data generation process, we have $I(Q; D_{\bar{T}}) = 0$ . The unflipped portion $D_{\bar{T}}$ contains no information about which concept the attacker selected as the target, because these labels were not modified and were generated independently of the attacker's choice.

For the flipped samples, we bound the mutual information under the worst-case assumption that each label flip can maximally exploit the information channel:

$$
\begin{aligned}
I(Q; D_T^{(p)} \mid D_{\bar{T}}) &\leq I(Q; D_T^{(p)}) \\
&= I(Q; \{(x_i, y_i')\}_{i \in T}) \\
&\leq \sum_{i \in T} I(Q; (x_i, y_i') \mid x_i) \\
&\leq \sum_{i \in T} \iota \\
&= |T| \cdot \iota \leq \epsilon N \cdot \iota
\end{aligned}
\tag{17}
$$

where the first inequality uses the fact that conditioning cannot increase mutual information, the third inequality applies the subadditivity of mutual information under the assumption that label flips are conditionally independent given features which represents a worst-case scenario for the defender, and the fourth inequality applies the per-sample bound established above.

Combining these results, we obtain:

$$I(Q; D^{(p)}) = I(Q; D_{\bar{T}}) + I(Q; D_T^{(p)} \mid D_{\bar{T}}) \leq 0 + \epsilon N \cdot \iota = \epsilon N \cdot \iota \tag{18}$$

This establishes the information injection constraint. The bound is tight when the attacker can use each flipped label to its full capacity. $\square$

### D.3.3 Combining Lower Bounds

**Corollary D.7.** *Combining Lemmas D.5 and D.6, for attack success rate* $\mathrm{ASR}(q_{k'}, \epsilon) \geq 1 - \delta$ *, we have:*

$$\epsilon \geq \frac{H(Q) - \log(1/\delta) - \log 2}{N \cdot \iota} \tag{19}$$

*Proof.* From Lemma D.5, achieving attack success rate $1 - \delta$ requires the mutual information to satisfy:

$$I(Q; D^{(p)}) \geq H(Q) - \log(1/\delta) - \log 2 \tag{20}$$

This represents the minimum information needed to reliably embed the backdoor.

From Lemma D.6, the maximum information that the attacker can inject through label flipping at rate $\epsilon$ is:

$$I(Q; D^{(p)}) \leq \epsilon N \cdot \iota \tag{21}$$

This represents the capacity of the label-flipping channel.

For the attack to succeed, we must have $\epsilon N \cdot \iota \geq H(Q) - \log(1/\delta) - \log 2$ . Rearranging yields the desired bound. $\square$

### D.4 Quantifying Attack Parameters

### D.4.1 Concept Entropy

**Lemma D.8** (Maximum Entropy)**.** *Under the uniform concept distribution assumption* $p_i = 1/K$ *for all* $i \in \{1, \ldots, K\}$ *, the concept entropy satisfies:*

$$H(Q) = \log K \tag{22}$$

*More generally, for any distribution over concepts, we have* $H(Q) \leq \log K$ *with equality if and only if the distribution is uniform.*

*Proof.* The entropy function achieves its maximum when the distribution is uniform (Raginsky et al., 2013), a fundamental property following from its concavity. For any probability distribution $(p_1, \ldots, p_K)$ with $\sum_{i=1}^{K} p_i = 1$ and $p_i \geq 0$ , we have:

$$H(Q) = -\sum_{i=1}^{K} p_i \log p_i \leq \log K \tag{23}$$

with equality if and only if $p_i = 1/K$ for all $i \in \{1, \ldots, K\}$ .

Under the uniform distribution assumption $p_i = 1/K$ for all $i$ , direct calculation yields:

$$\begin{aligned}
H(Q) &= -\sum_{i=1}^{K} p_i \log p_i \\
&= -\sum_{i=1}^{K} \frac{1}{K} \log \frac{1}{K} \\
&= -\frac{1}{K} \cdot K \cdot \log \frac{1}{K} \\
&= -\log \frac{1}{K} = \log K
\end{aligned} \tag{24}$$

In the $C^2$ attack setting, the attacker chooses which concept $q_{k'} \in \mathcal{C}$ to use as the trigger. From a defender's perspective without knowledge of the attacker's choice, the worst-case analysis assumes the attacker could

target any of the $K$ concepts with equal probability. This uniform prior maximizes uncertainty: $H(Q) = \log K$ , representing the maximum difficulty in defending against an unknown target concept. If the defender had prior knowledge that certain concepts are more likely to be targeted, then $H(Q) < \log K$ and the required flipping rate would be lower. The uniform distribution assumption is standard in minimax analysis and represents the hardest case for the defender. $\square$

### D.4.2 Per-Sample Information Budget

**Lemma D.9** (Information Budget per Flip). *The maximum information that can be injected per label flip satisfies:*

$$\iota \leq \log |\mathcal{Y}| \tag{25}$$

*where $|\mathcal{Y}|$ is the size of the label space, representing the number of classes in the classification task.*

*Proof.* This follows directly from the single-sample analysis in Lemma D.6's proof, but we provide additional context here.

Consider a single training example $(x_i, y_i)$ with feature $x_i$ and label $y_i \in \mathcal{Y}$ . The attacker observes $x_i$ and can choose to flip the label to any $y_i' \in \mathcal{Y}$ . The amount of information about concept $Q$ that can be embedded in the choice of $y_i'$ is bounded by:

$$I(Q; y_i' \mid x_i) \leq H(y_i' \mid x_i) \leq H(y_i') \leq \log |\mathcal{Y}| \tag{26}$$

where the first inequality follows from the data processing inequality (Raginsky et al., 2013) stating that $I(Q; Y) \leq H(Y)$ for any random variables $Q, Y$ , the second inequality uses the fact that conditioning reduces entropy as $H(Y|X) \leq H(Y)$ , and the third inequality applies the entropy bound for discrete random variables which states that for $Y$ taking $|\mathcal{Y}|$ values, $H(Y) \leq \log |\mathcal{Y}|$ with equality when $Y$ is uniform.

The bound $\iota = \log |\mathcal{Y}|$ is achieved when the attacker can freely choose any label $y_i' \in \mathcal{Y}$ and makes this choice to maximally encode information about $Q$ . In practice, the effective information per flip may be less than $\log |\mathcal{Y}|$ due to several factors: naturalness constraints requiring label flips to appear plausible, concept-label correlations where some concepts naturally correlate with certain labels thereby reducing the effective coding capacity, and detection risk where too many unnatural label flips may trigger anomaly detection. However, $\iota = \log |\mathcal{Y}|$ serves as a conservative upper bound representing the worst-case scenario for theoretical analysis, analogous to Shannon capacity in communication channels. $\square$

### D.4.3 Finite Sample Effects

**Lemma D.10** (Finite Sample Concentration). *With $N$ samples, the finite-sample correction satisfies $\Delta_N = \tilde{O}(\sqrt{\log K/N})$ . Specifically, with probability at least $1 - \delta$ :*

$$|H(\hat{Q}) - H(Q)| \leq \Delta_N = O\left(\sqrt{\frac{\log K \cdot \log(2/\delta)}{N}}\right) \tag{27}$$

*where $\hat{Q}$ is the empirical concept distribution and $H(Q)$ is the true population entropy.*

*Proof.* We establish concentration of the empirical entropy using McDiarmid's inequality (Combes, 2024) for functions with bounded differences. This technique is standard in robust statistics (Bhatt & Pensia, 2023).

Consider the empirical entropy as a function of $N$ independent and identically distributed samples $Q_1, \ldots, Q_N$ drawn from the concept distribution:

$$\hat{H}(\mathcal{C}) = -\sum_{i=1}^{K} \hat{p}_i \log \hat{p}_i \tag{28}$$

where $\hat{p}_i = \frac{1}{N} \sum_{j=1}^{N} \mathbf{1}[Q_j = i]$ is the empirical frequency of concept $i$ .

We first establish that changing a single sample $Q_j$ changes $\hat{H}(\mathcal{C})$ by at most $\frac{2\log K}{N}$ . Suppose we change $Q_j$ from concept $i$ to concept $i'$ . This changes $\hat{p}_i$ by $-1/N$ , $\hat{p}_{i'}$ by $+1/N$ , and leaves other $\hat{p}_k$ unchanged. The entropy is Lipschitz continuous with respect to the $\ell_1$ distance on the probability simplex: for any two distributions $\mathbf{p}, \mathbf{p}'$ on $K$ outcomes, we have $|H(\mathbf{p}) - H(\mathbf{p}')| \leq \|\mathbf{p} - \mathbf{p}'\|_1 \cdot \log K$ , which can be verified using the inequality $|x\log x - y\log y| \leq |x - y|\log K$ for $x, y \in [0, 1]$ . In our case, $\|\hat{\mathbf{p}} - \hat{\mathbf{p}}'\|_1 = 2/N$ , therefore:

$$|\hat{H}(\mathcal{C}) - \hat{H}(\mathcal{C})'| \leq \frac{2}{N} \cdot \log K = \frac{2\log K}{N} \tag{29}$$

This establishes the bounded difference property with $c_j = \frac{2\log K}{N}$ for all $j$ .

By McDiarmid's inequality (Combes, 2024), if a function $f(X_1, \ldots, X_N)$ satisfies the bounded difference property with constants $c_1, \ldots, c_N$ , then:

$$\mathbb{P}\left[|f - \mathbb{E}[f]| \geq t\right] \leq 2\exp\left(-\frac{2t^2}{\sum_{j=1}^N c_j^2}\right) \tag{30}$$

Applying this to $\hat{H}(\mathcal{C})$ with $c_j = \frac{2\log K}{N}$ yields:

$$\mathbb{P}\left[|\hat{H}(\mathcal{C}) - \mathbb{E}[\hat{H}(\mathcal{C})]| \geq t\right] \leq 2\exp\left(-\frac{Nt^2}{2(\log K)^2}\right) \tag{31}$$

Setting the right-hand side equal to $\delta$ and solving for $t$ gives:

$$t = \sqrt{\frac{2(\log K)^2\log(2/\delta)}{N}} = O\left(\log K\sqrt{\frac{\log(2/\delta)}{N}}\right) \tag{32}$$

By the law of large numbers, as $N \to \infty$ , the empirical entropy concentrates around the population entropy: $\mathbb{E}[\hat{H}(\mathcal{C})] \to H(\mathcal{C})$ . For finite $N$ , the bias $|\mathbb{E}[\hat{H}(\mathcal{C})] - H(\mathcal{C})|$ is at most $O(\log K/N)$ , which is negligible compared to the stochastic deviation when $N \gg \log K$ .

Using the triangle inequality to combine these results:

$$\begin{aligned}
|\hat{H}(\mathcal{C}) - H(\mathcal{C})| &\leq |\hat{H}(\mathcal{C}) - \mathbb{E}[\hat{H}(\mathcal{C})]| + |\mathbb{E}[\hat{H}(\mathcal{C})] - H(\mathcal{C})| \\
&\leq O\left(\log K\sqrt{\frac{\log(2/\delta)}{N}}\right) + O\left(\frac{\log K}{N}\right) \\
&= O\left(\sqrt{\frac{\log K \cdot \log(2/\delta)}{N}}\right)
\end{aligned} \tag{33}$$

with probability at least $1 - \delta$ , where the last step uses the fact that the $O(\log K/N)$ term is dominated by the $O\left(\sqrt{\log K/N}\right)$ term for $N \gg \log K$ . This establishes $\Delta_N = O\left(\sqrt{\frac{\log K \cdot \log(2/\delta)}{N}}\right)$ . $\qquad \square$

### D.5 Proof of Main Theorem

*Proof of Theorem D.1.* We combine the previous lemmas to establish the information-theoretic lower bound on the data flipping rate.

By Lemma D.5 applying Fano's inequality, any concept confusion attack achieving success rate ASR $\geq 1 - \delta$ must inject information satisfying:

$$I(Q; D^{(p)}) \geq H(Q) - \log(1/\delta) - \log 2 \tag{34}$$

This represents the fundamental information-theoretic barrier: to reliably cause misclassification with probability $1 - \delta$, the backdoor must encode at least $H(Q) - \log(1/\delta) - \log 2$ bits of information into the training data.

By Lemma D.6 on the information injection constraint, the maximum information that can be injected through label flipping at rate $\epsilon$ is:

$$I(Q; D^{(p)}) \leq \epsilon N \cdot \iota \tag{35}$$

This represents the communication capacity of the label-flipping channel: flipping $\epsilon N$ labels, each carrying at most $\iota$ bits, yields at most $\epsilon N \cdot \iota$ bits total.

For the attack to succeed, the available information must be sufficient to meet the minimum requirement. This is a necessary condition—if the attacker cannot inject enough information, then by Fano's inequality, it is information-theoretically impossible to achieve the target success rate. Combining the bounds established above:

$$\epsilon N \cdot \iota \geq H(Q) - \log(1/\delta) - \log 2 \tag{36}$$

Rearranging yields:

$$\epsilon \geq \frac{H(Q) - \log(1/\delta) - \log 2}{N \cdot \iota} \tag{37}$$

By Lemma D.8, under the uniform distribution assumption representing the worst-case scenario for the defender, $H(Q) = \log K$. By Lemma D.9, for an $|\mathcal{Y}|$-class task, $\iota \leq \log |\mathcal{Y}|$. Substituting these gives:

$$\epsilon \geq \frac{\log K - \log(1/\delta) - \log 2}{N \cdot \log |\mathcal{Y}|} \tag{38}$$

The analysis above assumes population-level quantities. For finite samples, Lemma D.10 shows that the empirical entropy concentrates: $|H(\hat{Q}) - H(Q)| = O(\sqrt{\log K/N})$. This introduces an additional correction term $\Delta_N$, but for $N \gg \log K$, this term is negligible compared to the main bound.

This completes the proof of Theorem D.1. $\qquad\square$

