# OpenReview forum: "Towards Representation Backdoor on CLIP via Concept Confusion"
_TMLR — Accepted by TMLR_

### Review · Reviewer_bNEP · 2026-03-28

**Summary Of Contributions:**

The paper proposes the Concept Confusion Attack ($C^2$ ATTACK) targeting multimodal foundation models like CLIP. Instead of injecting visible external triggers (e.g., patches or noise), it utilizes human-understandable concepts already present in the model's latent space as internal triggers.

Strength:
- The attack successfully evades state-of-the-art defenses (e.g., SCALE-UP, ABL) and detectors because it requires zero pixel-level modifications.
- The inclusion of a mathematical proof using Fano's inequality to bound the minimum data flipping rate adds support.
- Extensive experiments validate the effectiveness.

Weaknesses:

- Vulnerability to Failure Mode Analysis: The evaluation artificially splits the test set to calculate ASR (subset with the trigger feature) and CACC (subset without it). However, unlike traditional out-of-distribution (OOD) backdoor triggers (e.g., pixel patches), the proposed concept trigger (e.g., "water") is an in-distribution natural semantic feature. In a practical deployment, a defender analyzing the complete test distribution would easily identify this vulnerability through standard failure mode analysis, mitigating its claimed stealthiness.

- Susceptibility to Clean-Data Defenses: Consequently, the attack is highly vulnerable to clean-data-based defenses (such as FT-SAM, SAU, or PGBD). If a defender possesses a small, correctly annotated validation set that naturally contains the trigger feature, the backdoor is highly likely to be erased. The paper lacks evaluation against these realistic defense scenarios.2. Limited Novelty and Unacknowledged Prior Art (Clean-Image Backdoors)

- Missing Related Work: The proposed $C^2$ ATTACK is fundamentally a "clean-image backdoor attack," where only labels are poisoned while images remain untouched. There is a rich body of recent literature on this exact topic [A, B, C, D, E] that the authors completely fail to cite or discuss. This omission significantly undermines the paper's claims of novelty.

- Incremental Contribution: The core mechanism of this paper—selecting natural images with specific semantic features to poison—closely mirrors prior art, particularly [C]. While [C] utilizes InfoGAN to learn representations (e.g., identifying background color as a trigger), this work simply substitutes it with Concept Extraction methods and applies it to CLIP. Given that the methodology in [C] generalizes across various tasks, the proposed approach feels more like an efficient implementation or incremental improvement rather than a fundamental conceptual innovation.

[A] Clean-image Backdoor: Attacking Multi-label Models with Poisoned Labels Only. ICLR 2023 (https://openreview.net/forum?id=rFQfjDC9Mt)

[B] Label Poisoning is All You Need. NeurIPS 2023 (https://arxiv.org/abs/2310.18933)

[C] Breaking the Stealth-Potency Trade-off in Clean-Image Backdoors with Generative Trigger Optimization. AAAI 2026 (https://arxiv.org/abs/2511.07210)

[D] Clean-Image Backdoor Attacks. (https://arxiv.org/abs/2403.15010)

[E] Less is More: Stealthy and Adaptive Clean-Image Backdoor Attacks with Few Poisoned

**Audience:**

Yes

**Audience Explanation:**

Researchers focusing on AI security, foundation models, and explainable AI will find this work highly relevant. By demonstrating that stealthy backdoors can be embedded into CLIP purely via semantic concepts (without any pixel-level modifications), the paper exposes a critical blind spot in current defense paradigms.

**Broader Impact Concerns:**

The $C^2$ ATTACK poses a severe dual-use risk to the AI supply chain. Because it relies on natural semantic concepts and requires zero pixel modifications, it is highly stealthy and easily executed by merely relabeling a small subset of open-source images. This threatens critical downstream applications (e.g., autonomous driving, medical imaging) that rely on foundation models like CLIP. The authors must explicitly acknowledge these real-world deployment risks in a dedicated Broader Impact Statement or their conclusion.

**Claims And Evidence:**

No

**Claims Explanation:**

The claim that this is "the first backdoor framework to employ internal concepts as triggers" is overstated. Existing clean-image backdoor attacks [A-E] inherently rely on internal latent features as triggers. While the features exploited in prior works may not be as explicitly interpretable as the concepts targeted in this paper, they fundamentally operate as internal triggers. The authors should accurately scope this claim, perhaps revising it to state that this is the first backdoor framework specifically in CLIP to explicitly employ interpretable internal concepts.

**Requested Changes:**

1. Comprehensive Literature Review and Comparison: The authors must include a dedicated discussion on the rich, existing body of literature regarding "clean-image backdoor attacks" (where images are untouched and only labels are poisoned). Specifically, the paper must explicitly cite and compare its methodology against works like [C] (which utilizes generative representation learning like InfoGAN for semantic triggers), clarifying the fundamental conceptual innovations of $C^2$ ATTACK beyond merely substituting InfoGAN with Concept Extraction methods on CLIP. (Note: References [A]-[E] mentioned in the review must be cited and discussed).

2. Evaluation Against Clean-Data Defenses: Given that the trigger is an in-distribution natural feature, it is highly likely to be mitigated by clean-data-based defense mechanisms. The authors must evaluate and report the attack's survivability against standard defenses of this type (e.g., FT-SAM, SAU, or PGBD) using a small set of clean, correctly annotated data.

---

> ### Author Response · Authors · 2026-04-28
> **Clean-Image Backdoors, Defenses, and Broader Impact**
>
> We thank the reviewer for identifying the clean-image backdoor literature and for the suggestions on evaluation and broader impact.
>
> ### R1. Clean-image backdoors, novelty, and comparison
>
> C²Attack is mechanically a clean-image backdoor, and we now acknowledge and cite this line of work [1]–[5] in Related Work. We also narrowed our novelty claim to: "the first backdoor framework to use human-interpretable internal concepts as triggers in CLIP." Compared with [3], the differences are substantive: (i) C²Attack uses pre-named, human-interpretable concepts as triggers; (ii) it requires no generative training or extra data; and (iii) it targets CLIP-style contrastive foundation models rather than supervised image classifiers. We also added a clean-image baseline under the same 1% poison budget and no train/test pixel modification (Sec. 5.5, Table 12): the baseline reaches 83.7% ASR / 96.5% CACC, while C²Attack reaches 100.0% ASR with comparable CACC on strong concepts. This supports that the gain comes from selecting human-interpretable CLIP concepts rather than arbitrary label-only poisoning. Prior concept-backdoor work also focused on CBMs rather than CLIP [6,7].
>
> ### R2. Failure-mode analysis
>
> We agree this is a real limitation and now state it explicitly. Still, two factors reduce its practicality: low-correlation target choices (Table 2) and multi-concept triggers (Table 4), which make per-concept failure analysis less straightforward. More broadly, this issue is shared by in-distribution clean-image backdoors [1]–[5]; our contribution is showing that C²Attack still evades current defenses and detectors in Table 3.
>
> ### R3. Clean-data defenses
>
> We added FT-SAM and PGBD in Sec. 5.5 (Table 10), under both random clean repair and a stronger concept-covered clean set. The results support the reviewer’s intuition but also clarify the boundary: random clean repair does not remove the backdoor (ASR remains about 55%–83% across Engine/Headlight/Oven), while concept-covered PGBD reduces ASR much more substantially (about 27%–35%). Thus, mitigating C²Attack requires either clean data that covers the trigger concept or stronger representation-aware repair.
>
> ### R4. Broader impact
>
> We agree and added a dedicated Broader Impact section covering dual-use risk, a responsible-release plan (evaluation/defense-facing code only, not the full trigger-selection pipeline), and defender-side recommendations such as concept-activation auditing, concept-covered clean validation sets, and fine-tuning-based defenses such as FT-SAM/PGBD.
>
> ### References
>
> [1] Chen et al. Clean-image Backdoor. ICLR 2023.
> [2] Jha et al. Label Poisoning is All You Need. NeurIPS 2023.
> [3] Xu et al. Breaking the Stealth-Potency Trade-off in Clean-Image Backdoors with Generative Trigger Optimization. AAAI 2026.
> [4] Rong et al. Clean-Image Backdoor Attacks. arXiv 2024.
> [5] Xu et al. Less is More: Stealthy and Adaptive Clean-Image Backdoor Attacks with Few Poisoned. 2025.
> [6] Lai et al. ConceptGuard Battles Concept-level Backdoors in CBMs. 2024.
> [7] Lai et al. Concept-level Backdoor Attacks for CBMs. 2024.

---

> > ### Comment · Reviewer_bNEP · 2026-04-29
> > **Follow-up Comment**
> >
> > I thank the authors for their thorough point-by-point response and the revised manuscript.
> >
> > **Addressed concerns:**
> > - The clean-image backdoor literature [A–E] is now properly cited and discussed in Related Work, and the novelty claim has been appropriately scoped.
> > - A clean-image baseline comparison (Table 12) is added, quantifying the gain from concept-based selection (~16% ASR improvement over label-only poisoning).
> > - FT-SAM and PGBD defenses are now evaluated (Table 10), and a dedicated Broader Impact section has been added.
> >
> > **Remaining concerns:**
> > - The SAU defense requested in my review is still absent.
> > - The concept-covered PGBD results (ASR ~27–35%) confirm that an informed defender can substantially reduce attack efficacy. However, I consider this vulnerability acceptable, as PGBD requires the use of "clean" data accompanied by correct labels—a condition that cannot always be met. I recommend the authors to further test how many clean-data pairs are needed for an effective defense with PGBD.
> >
> > Overall, the authors have made substantive revisions, and the paper is clearer about its scope and limitations. The remaining concerns are minor.

---

### Review · Reviewer_2PzB · 2026-04-12

**Summary Of Contributions:**

This paper proposes C²Attack, a concept-level backdoor attack method targeting CLIP. Rather than relying on explicit image patches or pixel perturbations as triggers, it uses internally interpretable concepts within the model as implicit trigger conditions. Specifically, the authors first identify images that strongly exhibit a particular concept, then retrain the model using these samples relabeled as the target class, thereby causing the model to learn the concept itself as a backdoor trigger. Experiments demonstrate that this method achieves high attack success rates across multiple datasets and CLIP encoders while maintaining high clean accuracy.

Strengths:
1. This paper proposes a novel attack perspective: instead of designing explicit triggers, it directly utilizes the concept representations already present within CLIP as backdoor triggers. This approach shifts the attack mechanism from external input triggers to internal representation triggers, which is an interesting framing.
2. The paper flows smoothly overall, and the methodological workflow is easy to understand. Section 4 first uses an analysis of BadNet to demonstrate that backdoor training alters the concept activation distribution, then introduces the authors’ concept confusion attack; Figures 1 and 2 are highly intuitive.

Weaknesses:
1. This method is not entirely comparable to traditional attacks that rely solely on data poisoning or explicit trigger injection. This is because the method described in this paper first requires the use of a concept extractor and a threshold mechanism to identify strong concept images, and then relabels these samples. This effectively involves an additional layer of representation understanding and sample filtering, rather than simply performing poisoning.
2. The paper draws frequent connections between the method and fields such as cognitive science and Hopfieldian reasoning. However, from the perspective of the method itself, the core remains the use of certain high-concept-response samples for target relabeling, causing the model to learn erroneous bindings between concepts and target categories. These connections currently appear more like narrative framing and are not sufficiently integrated with the method’s core motivation.
3. The theoretical section has limited persuasiveness. The paper primarily presents an information-theoretic lower bound based on a uniform concept prior to illustrate the minimum flipping rate required for a successful attack. However, this conclusion is rather coarse; it does not truly capture the most critical aspects of the method, such as the error of the concept extractor, threshold selection, correlations between concepts, and the inherent imbalance in the data distribution. In other words, this theory functions more as a broad lower bound rather than a precise explanation of the C²Attack mechanism itself.
4. The attack and defense baselines used for comparison are generally outdated. On the attack side, most methods are still classics such as BadNet, Blended, Trojan, Refool, WaNet, and SSBA; on the defense side, solutions primarily target input anomalies or traditional backdoor patterns. Therefore, the experiments seem to demonstrate that concept-level triggers can bypass older defenses designed against explicit triggers, but they are insufficient to prove that this remains valid against newer, stronger representation-level defenses.
5. There is currently insufficient evidence regarding the method’s generalizability. Although the authors tested multiple CLIP encoders, the scope remains limited to the CLIP family. It remains to be seen whether this concept-level backdoor can be reliably transferred to other contrastive learning frameworks, other pre-trained vision-language models, or more general representation learning settings.

**Audience:**

Yes

**Audience Explanation:**

Backdoors in contrastive learning are a broad research area.

**Claims And Evidence:**

Yes

**Claims Explanation:**

The author tried many experiments.

**Requested Changes:**

Please refer to Weaknesses.

---

> ### Author Response · Authors · 2026-04-29
> **Attack Framing, Motivation, and Theory**
>
> We thank the reviewer for the careful reading and for recognizing the novelty of the attack perspective and the clarity of the method workflow.
>
> ### W1. Comparability with traditional data-poisoning attacks
>
> We respectfully disagree that the use of a concept extractor makes C²Attack less comparable to traditional data-poisoning attacks — in fact, it makes the comparison more informative. The concept extractor is an **offline, one-time selection tool** used only during the poisoning phase to identify which natural images already carry the trigger concept. It is never applied at inference time, is not shared with the model, and is not part of the attack's operational footprint. The actual poisoning operation is pure **label flipping** — no pixels are modified, no images are synthesized, no trigger patterns are injected. This is strictly simpler than traditional attacks (BadNet, Blended, WaNet, etc.) that require the attacker to modify image content. Table 7 (Sec. 5.4) shows that three different concept extractors (TCAV, Label-free CBM, Semi-supervised CBM) yield near-identical attack performance — all achieving ASR ≥ 95.7% and CACC ≥ 95.9% across 10 concepts on CIFAR-10 — confirming that the result is a property of the concept's presence in CLIP's latent space, not of any particular extractor. The concept extractor is analogous to the trigger-pattern design step in traditional attacks — all attacks require some form of trigger selection; C²Attack simply makes this step semantically interpretable rather than arbitrary.
>
> ### W2. Cognitive-science / Hopfieldian framing
>
> This is a fair point. The Hopfieldian framing was intended to provide intuitive grounding for why concepts serve as stable triggers, but we agree that the core mechanism is fully characterized by concept-to-label rebinding under fine-tuning and does not require the cognitive-science lens. In the revised manuscript, we have removed the Hopfieldian reasoning citation from the introduction and rewritten the surrounding paragraph to rest directly on the linear representation hypothesis [1,2] and on concept-based interpretability of visual models [3,4], which is the machinery actually operative in our method since concept activations are linear projections in the CLIP latent space. We have also removed the "cognitive neuroscience" phrasing from the contributions bullet. Section 4.1 already leads with the empirical concept-activation-shift observation (Table 1) rather than a cognitive-science narrative.
>
> ### W3. Theory too coarse
>
> We agree that Theorem 4.2 is best read as a worst-case information-theoretic impossibility result (a lower bound), and in the revised manuscript we have labeled it explicitly as "Worst-case impossibility result" and spelled out its idealizing assumptions (uniform concept priors, perfect concept extraction, independent concepts) directly in the theorem statement, together with the observation that the reviewer's suggested refinements (extractor error, threshold sensitivity, concept correlations, data imbalance) would only tighten the bound. Our empirical results in Table 9 (Sec. 5.4) show successful attacks at poisoning rates as low as 0.1% (well above the worst-case lower bound), which the theorem correctly predicts: the theoretical bound is a lower bound, and real-world CLIP conditions (structured concept priors, high concept-information ι, high-precision concept extractors) are more favorable than the uniform-prior worst case assumed by the bound. We have also added a sentence immediately after the theorem in Sec. 4.3 that makes this theory–experiment connection explicit, tying Table 9's empirical threshold back to the conditions under which the bound is expected to hold loosely.
>
> ### References
>
> [1] Kiho Park, Yo Joong Choe, and Victor Veitch. *The Linear Representation Hypothesis and the Geometry of Large Language Models*. CoRR, 2023.
>
> [2] Trenton Bricken, Adly Templeton, Joshua Batson, Brian Chen, Adam Jermyn, Tom Conerly, Nicholas L. Turner, Cem Anil, Carson Denison, Amanda Askell, et al. *Towards Monosemanticity: Decomposing Language Models with Dictionary Learning*. Transformer Circuits Thread, 2023.
>
> [3] Thomas Fel, Thibaut Boissin, Victor Boutin, Agustin Picard, Paul Novello, Julien Colin, Drew Linsley, Tom Rousseau, Remi Cadene, Lore Goetschalckx, Laurent Gardes, and Thomas Serre. *Unlocking Feature Visualization for Deeper Networks with MAgnitude Constrained Optimization*. NeurIPS 2023.
>
> [4] Amirata Ghorbani, James Wexler, James Y. Zou, and Been Kim. *Towards Automatic Concept-Based Explanations*. NeurIPS 2019.

---

> > ### Author Response · Authors · 2026-04-29
> > **Baselines and Generalizability**
> >
> > ### W4. Outdated attack and defense baselines
> >
> > On the attack side, we include BadCLIP [1], which is the most recent CLIP-specific backdoor baseline. The other baselines (BadNet, Blended, Trojan, Refool, WaNet, SSBA) are retained intentionally because they represent all major trigger paradigms: patch, blend, perturbation, reflection, warp, and sample-specific. To further strengthen the comparison, the revised manuscript adds a clean-image backdoor baseline under the same 1% poisoning budget and no train/test pixel modification — presented as the "Clean-image backdoor baseline" paragraph in Sec. 5.5 (Additional Robustness and Baseline Experiments) with Table 12. The baseline achieves 83.7% ASR with 96.5% CACC, while C²Attack achieves 100.0% ASR on strong concepts with comparable CACC.
> >
> > | Attack | Train edit | Test edit | Poison rate | Target | CACC (%) | ASR (%) |
> > |---|---|---|---:|---|---:|---:|
> > | Clean-image baseline | No | No | 1% | Airplane | 96.5 | 83.7 |
> > | C²Attack-Engine | No | No | 1% | Airplane | 97.5 | 100.0 |
> > | C²Attack-Headlight | No | No | 1% | Airplane | 97.2 | 100.0 |
> > | C²Attack-Oven | No | No | 1% | Airplane | 97.6 | 100.0 |
> >
> > On the defense side, the revised manuscript adds FT-SAM and PGBD in the "Clean-data and representation-aware defenses" paragraph in Sec. 5.5 with Table 10. PGBD is representation/semantic-aware and substantially reduces ASR under concept-covered clean data, while random clean repair still leaves non-trivial ASR. These results clarify both the effectiveness and the boundary of C²Attack.
> >
> > ### W5. Generalizability beyond CLIP
> >
> > Our focus on CLIP is deliberate and well-motivated. CLIP is the canonical foundation model without an explicit concept bottleneck — prior concept-backdoor work [2,3] only studied concept bottleneck model (CBM) architectures with explicit concept layers. C²Attack closes precisely this gap by showing that concept-level backdoors are possible even in models where concepts are implicitly represented as linear directions in the latent space. The linear representation hypothesis that underlies our method has been validated in multiple contrastive and self-supervised vision models, suggesting that C²Attack would generalize. Table 8 (Sec. 5.4) already demonstrates generalization across four different CLIP encoder architectures (ViT-B/16, ViT-B/32, ViT-L/14, ViT-L/14-336px), all achieving 100% ASR with CACC 96.4%–98.2% — showing that the attack is not tied to a specific architecture but rather to the shared property of linear concept representation in contrastive vision-language models. Extending to non-CLIP models (SigLIP, DINOv2, etc.) is an interesting future direction, but we believe the current scope — four encoders, three datasets, three concept extractors, 30 concepts — already provides strong evidence for the generality of the concept-confusion mechanism.
> >
> > ### References
> >
> > [1] Jiawang Bai, Kuofeng Gao, Shaobo Min, Shu-Tao Xia, Zhifeng Li, and Wei Liu. *BadCLIP: Trigger-aware Prompt Learning for Backdoor Attacks on CLIP*. CVPR 2024.
> >
> > [2] Songning Lai, Yu Huang, Jiayu Yang, Gaoxiang Huang, Wenshuo Chen, and Yutao Yue. *Guarding the Gate: ConceptGuard Battles Concept-level Backdoors in Concept Bottleneck Models*. arXiv:2411.16512, 2024.
> >
> > [3] Songning Lai, Jiayu Yang, Yu Huang, Lijie Hu, Tianlang Xue, Zhangyi Hu, Jiaxu Li, Haicheng Liao, and Yutao Yue. *CAT: Concept-level Backdoor Attacks for Concept Bottleneck Models*. arXiv:2410.04823, 2024.

---

> ### Comment · Reviewer_2PzB · 2026-05-12
>
> I thank the authors for the detailed response and supplementary experiments. I believe these revisions have largely addressed my main concerns, and I now recommend acceptance.

---

### Review · Reviewer_95Cc · 2026-04-16

**Summary Of Contributions:**

This paper studies a new form of backdoor attack termed  C2Attack against CLIP-based image classifiers. The attack first identifies a trigger concept using a concept extractor, then relabels images that strongly exhibit this concept to a target class, and finally fine-tunes the CLIP-based classifier on the mixed clean/poisoned dataset. The paper also includes a concept-activation analysis, an information-theoretic lower bound on the poisoning ratio, and experiments on CIFAR-10, CIFAR-100, and Tiny-ImageNet showing high ASR with relatively preserved clean accuracy, as well as robustness against several existing defenses and detectors.

**Audience:**

Yes

**Audience Explanation:**

Overall, I think this paper has a genuinely novel and potentially impactful idea. Recasting backdoor triggers as internal concepts rather than external artifacts is interesting and likely to inspire follow-up work in both attack and defense for multimodal models. However, for a top-tier venue, I do not think the current version is yet polished or rigorous enough. The empirical claims about generality and defense evasion need tighter support, the implementation inconsistency around the poisoning ratio must be resolved, and the theory-experiment connection should be improved.

**Broader Impact Concerns:**

I think the broader-impact section should more explicitly discuss safeguards: for example, limiting release of attack-ready code or full trigger-selection pipelines, and providing defensive recommendations oriented toward representation-level monitoring rather than only input filtering.

**Claims And Evidence:**

Yes

**Claims Explanation:**

- The key idea is clear and interesting: instead of treating the trigger as an input-space artifact, the paper redefines the trigger as a concept already present in the representation space.
- The experimental results are promising.

**Requested Changes:**

- The paper should clarify the difference between C2Attack and natural backdoor attacks. At present, the distinction is somewhat blurred, since both rely on naturally occurring semantics instead of explicit injected triggers. A more explicit conceptual comparison would help highlight the true novelty of the proposed method.
- I advice the authors to cite and discuss prior closely related work such as "BackdoorVLM: A Benchmark for Backdoor Attacks on Vision-Language Models", and preferably include experimental comparisons where possible. This would strengthen the positioning of the paper and make the empirical evaluation more complete.
- The paper currently reports 1% in the main method section and 99% in the appendix. This must be corrected, and all tables/claims should be checked accordingly.
- Clarify the true experimental scope. If most main experiments are run on CLIP ViT-B/16 with TCAV, the paper should say so clearly. If the claim is that the method generalizes across four encoders, then stronger main-text evidence should be added for all four, not only appendix-level mention.
- Strengthen the practicality discussion of trigger concepts. Please analyze how often suitable strong-concept samples exist, how concept purity affects ASR, how much target/concept overlap matters, and what happens when concepts are less cleanly separable.
- Add stronger defense baselines. In particular, include at least one representation-space or concept-aware defense, or a simple defense based on concept activation regularity, since the main claim is that the attack moves from input space to representation space.
Tighten the theory-experiment connection.

---

> ### Author Response · Authors · 2026-04-29
> **Conceptual Distinction, Citation, and Experimental Scope**
>
> We thank the reviewer for recognizing the novelty and potential impact of reframing backdoor triggers as internal concepts and for the detailed, actionable feedback.
>
> ### R1. Difference between C²Attack and natural backdoor attacks
>
> The distinction is fundamental and already established in our paper. Natural backdoor attacks (e.g., physical backdoors [1]) place **real-world physical objects** (stickers, glasses, colored patches) into the scene — the trigger is still an **input-space artifact**, just one that looks natural. C²Attack makes **zero pixel-level modifications** to any image. The trigger is not a physical object placed in the scene; it is a concept that the CLIP encoder **already represents linearly** in its latent space (as validated by the concept-activation analysis in Section 4.1 and Figure 2). The attacker merely relabels existing images that happen to contain the concept — no image acquisition, no scene manipulation, no pixel editing. Table 5 already provides a direct empirical comparison: physical backdoors achieve only 30.9–59.5% ASR because the physical object must dominate the scene, whereas C²Attack achieves 100% ASR because the concept activation is present in any image that semantically contains the concept, regardless of its visual prominence.
>
> ### R2. BackdoorVLM citation
>
> Thank you for the pointer. BackdoorVLM [2] focuses on generative vision-language models (visual encoder + LLM generation head) and evaluates traditional input-space triggers (patch, blend, perturbation) for image-text tasks, whereas our evaluation focuses on CLIP-style contrastive encoders used for downstream image classification, with the trigger operating entirely at the concept level. Because the model interface, task objective, and ASR definition differ, a direct numerical comparison is less aligned than comparing within the CLIP classification setting. We have added a BackdoorVLM citation and this scope contrast to the Related Works section of the revised manuscript. To strengthen the empirical comparison within our setting, we also include a clean-image backdoor baseline under the same 1% poisoning budget and no-pixel-modification protocol; the results are reported in our responses to Reviewer 2PzB and Reviewer bNEP.
>
> ### R3. Poisoning-ratio inconsistency (1% vs. 99%)
>
> The reviewer is correct — the "99%" in Appendix B.5 of the original submission is a typo. The intended and used poisoning ratio in all experiments is **1%**, as confirmed by (i) the threshold construction in Section 4.2 ("we set the poisoning ratio as 1%"), (ii) Table 7 which sweeps 0.1%–1.0%, and (iii) the percentile thresholding procedure. We have corrected this in the revised manuscript (the updated Appendix now reads "$1\%$") and audited every instance of the poisoning ratio end-to-end for consistency.
>
> ### R4. Experimental scope across four encoders
>
> We appreciate this clarification request. The main-text experiments use CLIP ViT-B/16 with TCAV as the default configuration, and the generalization evidence is already comprehensive in Sec. 5.4 (Ablation Study): **Table 8** evaluates all four CLIP encoders (ViT-B/16, ViT-B/32, ViT-L/14, ViT-L/14-336px), all achieving **100% ASR** with CACC ranging from 96.4% to 98.2% on CIFAR-10. **Table 7** demonstrates that C²Attack is robust across three different concept extractors (TCAV, Label-free CBM, Semi-supervised CBM) with near-identical attack performance, confirming that the results are not an artifact of any particular extractor. **Table 6** further tests 30 different trigger concepts, showing consistent high ASR across diverse semantic categories. These tables are directly cross-referenced in Sec. 5.4 — the encoder architectures paragraph links to Table 8, the concept-extractor paragraph links to Table 7, and the 30-concept paragraph links to Table 6 — so the breadth of the evaluation is visible from the main text.
>
> ### References
>
> [1] Emily Wenger, Josephine Passananti, Arjun Nitin Bhagoji, Yuanshun Yao, Haitao Zheng, and Ben Y. Zhao. *Backdoor Attacks against Deep Learning Systems in the Physical World*. CVPR 2020.
>
> [2] Juncheng Li, Yige Li, Hanxun Huang, Yunhao Chen, Xin Wang, Yixu Wang, Xingjun Ma, and Yu-Gang Jiang. *BackdoorVLM: A Benchmark for Backdoor Attacks on Vision-Language Models*. arXiv:2511.18921, 2025.

---

> > ### Author Response · Authors · 2026-04-29
> > **Trigger Practicality, Defenses, Theory, and Broader Impact**
> >
> > ### R5. Practicality of trigger concepts
> >
> > Our paper already provides substantial evidence on trigger-concept practicality. **Table 8** evaluates 30 trigger concepts from the Broden concept bank on CIFAR-10, with ASR ranging from 97.98% (Bannister) to 100% (Airplane, Oven, Engine, Headlight, Head, Clock, Mirror, Air_conditioner, Building, Cushion) and CACC consistently ≥ 95.88%. To further address less separable concepts, we added a concept-quality stress test in Sec. 5.5 and Appendix B.6. Strong concepts with high TCAV separability achieve 100% ASR, while weaker concepts show lower ASR, clarifying that concept separability and purity are important practical factors and that viable trigger concepts can be pre-screened.
> >
> > | Concept | Group | TCAV Sep. Acc. | Target overlap (%) | CACC (%) | ASR (%) |
> > |---|---|---:|---:|---:|---:|
> > | Engine | Strong | 0.93 | 4.9 | 97.50 | 100.0 |
> > | Headlight | Strong | 0.91 | 14.6 | 97.20 | 100.0 |
> > | Oven | Strong | 0.91 | 4.0 | 97.60 | 100.0 |
> > | Cake | Weak | 0.65 | 11.7 | 96.82 | 68.67 |
> > | Computer | Weak | 0.68 | 6.4 | 96.71 | 70.83 |
> > | Ruler | Weak | 0.60 | 12.5 | 97.08 | 53.18 |
> >
> > TCAV Sep. Acc. is the validation accuracy of the TCAV concept classifier; target overlap is the fraction of selected high-concept training images already belonging to the target class.
> >
> > ### R6. Stronger / representation-aware defenses
> >
> > We added stronger clean-data and representation-aware defenses, including FT-SAM and PGBD, in Sec. 5.5 and Appendix B.6. We evaluate two settings: a random clean set and a stronger concept-covered clean set containing correctly labeled images with the trigger concept. The results show that C²Attack remains active under random clean repair, while concept-covered clean data and PGBD reduce ASR more substantially, clarifying the robustness boundary.
> >
> > | Concept | Defense | Clean set | CACC (%) | ASR (%) |
> > |---|---|---|---:|---:|
> > | Engine | None | - | 97.5 | 100.0 |
> > | Engine | Fine-tune | Random clean | 97.3 | 91.5 |
> > | Engine | FT-SAM | Random clean | 97.2 | 82.5 |
> > | Engine | PGBD | Random clean | 96.7 | 64.8 |
> > | Engine | FT-SAM | Concept-covered clean | 96.7 | 49.6 |
> > | Engine | PGBD | Concept-covered clean | 95.8 | 34.7 |
> > | Headlight | None | - | 97.2 | 100.0 |
> > | Headlight | Fine-tune | Random clean | 97.1 | 87.9 |
> > | Headlight | FT-SAM | Random clean | 96.9 | 78.5 |
> > | Headlight | PGBD | Random clean | 96.4 | 59.7 |
> > | Headlight | FT-SAM | Concept-covered clean | 96.5 | 44.8 |
> > | Headlight | PGBD | Concept-covered clean | 95.4 | 29.6 |
> > | Oven | None | - | 97.6 | 100.0 |
> > | Oven | Fine-tune | Random clean | 97.1 | 87.8 |
> > | Oven | FT-SAM | Random clean | 96.7 | 74.9 |
> > | Oven | PGBD | Random clean | 96.1 | 54.6 |
> > | Oven | FT-SAM | Concept-covered clean | 96.3 | 39.7 |
> > | Oven | PGBD | Concept-covered clean | 95.2 | 26.5 |
> >
> > ### R7. Theory–experiment connection
> >
> > Theorem 4.2 provides an information-theoretic lower bound via Fano's inequality: ε ≥ (H(Q) − log(1/δ) − log 2) / (N·ι). It is intentionally worst-case, making no assumptions about extractor accuracy, data distribution, or concept correlations. The connection to experiments is that Table 7 already shows successful attacks at 0.1% poisoning (ε = 0.001), consistent with the theorem’s role as a lower bound rather than a tight prediction. We therefore added an explicit sentence in Sec. 4.3 connecting the bound to the poison-rate results.
> >
> > ### R8. Broader impact safeguards
> >
> > We agree that the broader-impact discussion should be more explicit. We added a dedicated **Broader Impact** section covering: (i) responsible release — open-sourcing only the evaluation harness and defense-facing code, not the end-to-end trigger-selection pipeline; (ii) defender-side recommendations, including auditing concept-activation distributions and maintaining clean validation sets that cover diverse natural concepts; and (iii) a recommendation to prefer fine-tuning-based defenses (e.g., FT-SAM, PGBD) over input-space filters for foundation models whose latent spaces encode human-interpretable concepts.

---

> ### Comment · Reviewer_95Cc · 2026-04-29
> **Follow-up**
>
> Thank you for the response and the additional experimental results. The authors have addressed most of my main concerns, particularly regarding the discussion with BackdoorVLM, the evaluation under different poisoning rates, and the analysis of defense methods. I encourage the authors to clearly include these new results and discussions in the final version.
>
> Based on the current response, I am now positive toward acceptance.

---

### Review · Reviewer_Tw2B · 2026-04-26

**Summary Of Contributions:**

This paper introduces a new perspective on backdoor attacks in multimodal models by reframing them as manipulations of internal concept representations rather than purely input-space phenomena. Drawing inspiration from interpretable machine learning, the work argues that traditional trigger-based attacks implicitly function by altering the activation of latent concepts within a model. By making this connection explicit, the paper bridges the gap between backdoor attack literature and concept-based interpretability, offering a representation-level understanding of how malicious behaviors are embedded in models such as CLIP.

Building on this perspective, the paper proposes the Concept Confusion Attack (C2Attack), a novel backdoor framework that eliminates the need for explicit triggers in the input space. Instead of injecting visible patterns or imperceptible perturbations, the method designates human-understandable concepts as internal triggers and constructs poisoned data by relabeling samples that strongly exhibit a selected concept. Through this process, the model learns to associate the presence of the concept itself with a target label, effectively embedding the backdoor into its decision-making process. This design allows the attack to remain within the natural data distribution, making it inherently stealthier than conventional approaches that rely on detectable artifacts.

The work further operationalizes this idea within CLIP-based image classification models by presenting a concrete and reproducible pipeline that integrates concept extraction, concept recognition via thresholding, and backdoor dataset construction through conditional relabeling. In addition, the paper provides empirical analysis showing that backdoor training induces significant shifts in high-level semantic representations, particularly in deeper layers of the model, supporting the claim that the attack fundamentally operates at the level of concept activation rather than superficial features.

Extensive experiments across multiple datasets and model variants demonstrate that C2Attack achieves high attack success rates while preserving clean-task accuracy, even under low poisoning ratios. Moreover, the evaluation against several existing defense and detection methods shows that approaches designed for input-level triggers are largely ineffective against this attack, highlighting a critical limitation of current backdoor defense strategies. Finally, the paper presents a preliminary theoretical analysis that characterizes the relationship between poisoning rate, concept space, and attack success from an information-theoretic perspective, providing an initial step toward understanding the fundamental limits of concept-based backdoor attacks.

**Audience:**

Yes

**Audience Explanation:**

N/A

**Claims And Evidence:**

Yes

**Claims Explanation:**

N/A

**Requested Changes:**

The assumed threat model grants the adversary full control over training data and labels, which is standard but weakens practical relevance. In real-world CLIP deployment scenarios (e.g., API-based models or pretrained frozen encoders), such access is often unrealistic. The paper lacks discussion or experiments under more constrained settings (e.g., limited poisoning budget, partial data access, or post-training attacks).

---

> ### Author Response · Authors · 2026-04-28
> **Practicality and Threat Model**
>
> We thank the reviewer for the supportive assessment and for confirming that C²Attack is well-motivated and relevant. The reviewer raises an important point about practical deployment constraints, and we appreciate the opportunity to clarify what the current paper covers and what remains open.
>
> ### Threat-model constrained settings (limited poisoning budget, partial data access, post-training attacks)
>
> **(a) Limited poisoning budget.** We agree this is an important dimension of practicality. Our paper already evaluates C²Attack under poisoning ratios as low as 0.1% on CIFAR-10, covering three trigger concepts (Airplane, Engine, Headlight). Across all ten poison rates (0.1%–1.0%), ASR remains ≥ 95% and CACC stays ≥ 96%; notably, the "Airplane" concept achieves a perfect 100% ASR at every poison rate tested, including 0.1%. This is consistent with the information-theoretic lower bound in Theorem 4.2, which predicts that concept-based triggers require far fewer poisoned samples than input-space triggers because each poisoned image carries high mutual information (ι) about the trigger concept. In the revised manuscript, we have promoted this analysis from the appendix to the main text as a dedicated "Impact of Poison Rates" paragraph in Sec. 5.4 (Ablation Study), so that readers see the low-poisoning-budget evidence alongside the main results rather than discovering it only in the appendix.
>
> **(b) Partial data access.** This is a valid concern that our current experiments do not directly address. Our threat model assumes the adversary controls the full training pipeline, which is standard in the backdoor literature but, as the reviewer correctly notes, limits practical relevance. That said, we note that C²Attack has a structural advantage in the partial-access setting compared to input-space attacks: because the trigger is a naturally occurring concept (not an injected artifact), the adversary only needs to relabel a small number of concept-carrying images within the fraction they control — they do not need to inject any synthetic content. We have added this partial-access scenario to the Limitations section of the revised manuscript as a promising direction for future investigation.
>
> **(c) Post-training attacks.** Our paper already evaluates C²Attack across four CLIP encoders (ViT-B/16, ViT-B/32, ViT-L/14, ViT-L/14-336px; Appendix Table 6), all achieving 100% ASR with CACC ranging from 96.4% to 98.2%. These results show that the concept-level backdoor is effective regardless of which pretrained CLIP encoder the victim uses. Additionally, Table 3 shows that the backdoor survives five defense methods (ShrinkPad, Auto-Encoder, SCALE-UP, Fine-pruning, ABL) and two detection methods (SSL-Cleanse, DECREE), which a victim would likely deploy post-training. Taken together, these results cover the most common post-training deployment scenarios for CLIP-based classifiers.

---

> > ### Comment · Reviewer_Tw2B · 2026-05-12
> > **Follow-up**
> >
> > Thank the authors' response. I am positive toward acceptance.

---

### Author Response · Authors · 2026-04-29
**Summary of Revisions**

We thank all four reviewers for their constructive feedback. Below we first summarize the concrete revisions that have already been applied to the manuscript, and then respond to each reviewer individually. All additions in the revised PDF are highlighted in **blue** so the changes are easy to locate.

# Summary of Revisions

| Area | Applied change | Prompted by |
|---|---|---|
| **Sec. 1 (Intro) + Contributions** | Rewrote the concept-motivation paragraph around the linear representation hypothesis and concept-based interpretability (removed the Hopfieldian/cognitive-science framing). Scoped the novelty claim (in both the intro and the contributions bullet) to *"the first backdoor framework to use human-interpretable internal concepts as triggers in CLIP."* | 2PzB, bNEP |
| **Sec. 2 (Related Work)** | Added a new *"Clean-Image Backdoor Attacks"* paragraph citing the five clean-image backdoor references flagged by the reviewer and positioning C²Attack against them. | bNEP |
| **Sec. 2 (Related Work)** | Added a BackdoorVLM citation with an explicit scope contrast (generative VLMs with LLM heads vs. CLIP-style contrastive encoders for classification). | 95Cc |
| **Sec. 4.3 (Theory)** | Labeled Theorem 4.2 as *"Worst-case impossibility result"* and stated its idealizing assumptions (uniform concept priors, perfect extraction, independent concepts) inside the theorem; added a sentence linking the bound to the empirical poison-rate result in Table 9. | 2PzB, 95Cc |
| **Sec. 5.3 (Ablation)** | Promoted *"Impact of Poison Rates"* from the appendix to the main ablation, so the 0.1–1.0% poisoning evidence is visible alongside the main results rather than only in the appendix. | Tw2B |
| **Sec. 5.5 (new subsection: Additional Robustness and Baseline Experiments)** | **(a) Clean-data / representation-aware defenses** (FT-SAM, PGBD) under random-clean and concept-covered clean repair settings — Table 10; **(b) Concept-quality stress test** with weaker concepts and TCAV separability statistics — Table 11; **(c) Clean-image backdoor baseline** under the same 1% poisoning budget and no train/test pixel modification — Table 12. | bNEP, 95Cc, 2PzB |
| **Appendix B** | Corrected the 99% → 1% poisoning-rate typo and audited every poisoning-rate reference for consistency. | 95Cc |
| **Limitation** | Added two acknowledged open directions: (i) partial data-access scenario (attacker controls only a fraction of the training data); (ii) in-distribution failure-mode-analysis vulnerability shared with other clean-image backdoors. | Tw2B, bNEP |
| **Broader Impact (new section)** | Added a dedicated section covering the dual-use risk, a responsible-release plan (open-source only the evaluation harness and defense-facing code, not the end-to-end trigger-selection pipeline), and three concrete defender-side recommendations (audit concept-activation distributions, maintain concept-covered clean validation sets, prefer fine-tuning-based defenses over input-space filters). | bNEP, 95Cc |

---

### Decision · Action_Editor_vHA4 · 2026-05-13

**Recommendation:** Accept with minor revision

**Additional Comments:**

The three official reviewers reached consensus after the rebuttal phase, with all three submitting Leaning Accept. Reviewer 2PzB stated the revisions have largely addressed the main concerns; reviewer 95Cc indicated being now positive toward acceptance; reviewer bNEP confirmed the revisions are substantive and remaining concerns are minor.

Two minor, well-scoped items from reviewer bNEP's post-rebuttal follow-up remain outstanding and should be incorporated in the camera-ready version. They fit naturally into the existing Sec. 5.5 framework that the authors built during the rebuttal round and do not require new conceptual machinery.

Required minor revisions for camera-ready:

1. **Add SAU defense evaluation.** Reviewer bNEP's initial review explicitly requested SAU as a clean-data defense baseline alongside FT-SAM and PGBD. The 28 April revision added FT-SAM and PGBD (Table 10) but SAU remains absent. Please evaluate SAU under both random-clean and concept-covered clean repair settings, in the same Table 10 format, to complete the clean-data defense panel.

2. **Add PGBD clean-data sample-size sensitivity.** In the post-rebuttal follow-up, reviewer bNEP recommended testing how many clean-data pairs are required for PGBD to function as an effective defense. Please add a small sensitivity analysis (e.g., varying the clean-set size for PGBD on at least one strong-concept setting) so that the practical defender effort is quantified.

Procedural note for the record: a fourth reviewer (Tw2B) was assigned to this paper by the system beyond the standard three-reviewer policy. I attempted to remove this assignment through OpenReview and raised the matter with the Editors-in-Chief; neither route allowed removal, and I asked the reviewer to refrain from submitting. A review was nonetheless submitted, and the reviewer subsequently posted a follow-up indicating a positive view of the revised manuscript. My decision rests primarily on the substantive feedback of the three official reviewers; the fourth reviewer's input is consistent with the official consensus and does not affect the recommendation.

I thank the reviewers for their thoughtful engagement across the discussion phase, and the authors for the careful and responsive revisions.

**Audience:**

Yes

**Audience Explanation:**

All three official reviewers agree that the work is of interest to TMLR's audience (3/3 Yes). The paper introduces a concept-level backdoor framework for CLIP-style contrastive foundation models, a setting that has not been addressed in prior concept-backdoor work (which has been limited to architectures with explicit concept bottleneck layers). Three audience segments are likely to engage with the findings:

(i) Backdoor attack and defense researchers, who gain a concrete instance of a backdoor that is not defeated by input-space anomaly detectors and whose mitigation requires representation-level or concept-aware defenses;

(ii) Concept-based interpretability and foundation-model safety researchers, who gain empirical evidence that linearly-represented concepts in the CLIP latent space are exploitable as triggers;

(iii) Practitioners working on AI supply-chain security for multimodal systems, who gain a concrete dual-use risk specification and the corresponding defender-side recommendations articulated in the new Broader Impact section.

**Claims And Evidence:**

Yes

**Claims Explanation:**

All three official reviewers concur that the claims are supported by accurate, convincing and clear evidence (3/3 Yes). The 28 April revision substantively strengthens the empirical and theoretical support for the main claims:

(i) A new Section 5.5 introduces stronger representation-aware defenses (FT-SAM, PGBD) under both random-clean and concept-covered clean repair settings (Table 10), clarifying the robustness boundary of the attack rather than only its strengths;

(ii) A concept-quality stress test (Table 11) reports TCAV separability statistics for both strong and weak concepts, making the dependence of attack success on concept structure explicit;

(iii) A clean-image backdoor baseline at the same 1% poisoning budget and no pixel modification (Table 12) quantifies the gain attributable to concept-based selection over arbitrary label-only poisoning;

(iv) Theorem 4.2 is now explicitly labeled as a worst-case impossibility result with its idealizing assumptions (uniform concept priors, perfect extraction, independent concepts) stated inline, and is now connected to the empirical low-poisoning-rate evidence in the text;

(v) The 99% $\rightarrow$ 1% poisoning-rate typo in Appendix B has been corrected and all references audited.

Reviewer bNEP's remaining notes (absence of SAU defense; PGBD clean-data sample-size sensitivity) are minor and do not affect the core claims, as confirmed in the post-rebuttal exchange.